# Revisiting Neural Networks for Few-Shot Learning: A Zero-Cost NAS Perspective

**Haidong Kang** [1]

## Abstract

Neural Architecture Search (NAS) has recently outperformed hand-designed networks in various artificial intelligence areas. However, previous works only target a pre-defined task. For a new task in few-shot learning (FSL) scenarios, the architecture is either searched from scratch, which is neither efficient nor flexible, or borrowed architecture from the ones obtained on other tasks, which may lead to sub-optimal. Can we select the best neural architectures without involving any training and eliminate a significant portion of the search cost for new tasks in FSL? In this work, we provide an affirmative answer by proposing a novel information bottleneck (IB) theory-driven *Few-shot Neural Architecture Search* (dubbed, IBFS) framework to address this issue. We first derive that the global convergence of Model-agnostic meta-learning (MAML) can be guaranteed by only considering the first-order loss landscape. Moreover, motivated by the observation that IB provides a unified view toward understanding machine learning models, we propose a novel Zero-Cost method tailored for FSL to rank and select architectures based on their *expressivity* obtained by IB mechanisms. Extensive experiments show that IBFS achieves state-of-the-art performance in FSL without training, which demonstrates the effectiveness of our IBFS.

## 1. Introduction

Deep Neural Networks (DNNs) have shown remarkable performance on various challenging real-world tasks, i.e., image recognition (He et al., 2016), and few-shot learning (Finn et al., 2017; Sun & Gao, 2024). However, this

[1]College of Software, Northeastern University, Shenyang, China. Correspondence to: Haidong Kang <hdkang@stumail.neu.edu.cn>.

*Proceedings of the $42^{nd}$ International Conference on Machine Learning*, Vancouver, Canada. PMLR 267, 2025. Copyright 2025 by the author(s).

**The best directions:** ⟶
The less costs (search/train/memory) the better, and the higher accuracy and generalization the better.

IBFS
AutoMeta
MAML
Vanilla NAS

Costs (search/train/memory)   Generalization   Accuracy

*Figure 1.* **IBFS** *vs.* **Peer competitors** in terms of costs, accuracy, and generalization.

raises an important concern about whether the frameworks of the DNNs are ideal for those tasks. Naturally, NAS (Liu et al., 2019b; Zoph et al., 2018) is a straightforward solution for exploring frameworks for those tasks (i.e., FSL) due to its automated search manner, which can automatically search high-performance architectures for the aforementioned tasks.

**Problem statement.** Though promising, conventional NAS has *two fundamental limitations*. First, conventional NAS methods only focus on a single task. As shown in Fig. 2a, vanilla NAS only targets a predefined target, for a new task, the architecture is either searched from scratch, which is time-consuming, or borrowed from the ones obtained on other tasks, which may lead to sub-optimal. Second, as depicted in Fig. 2b, despite adaption NAS (Kim et al., 2018; Lian et al., 2020; Elsken et al., 2020) based on meta-learning can enhance the generalization of searched architecture on unseen tasks, however, it is extremely time-consuming, requiring over 100 GPU days (Kim et al., 2018) for searching.

**Our goal.** In this paper, we aim to address the above two fundamental limitations by revisiting neural networks tailored for FSL from a zero-shot NAS perspective. As shown in Fig. 1, our IBFS achieves better performance in terms of

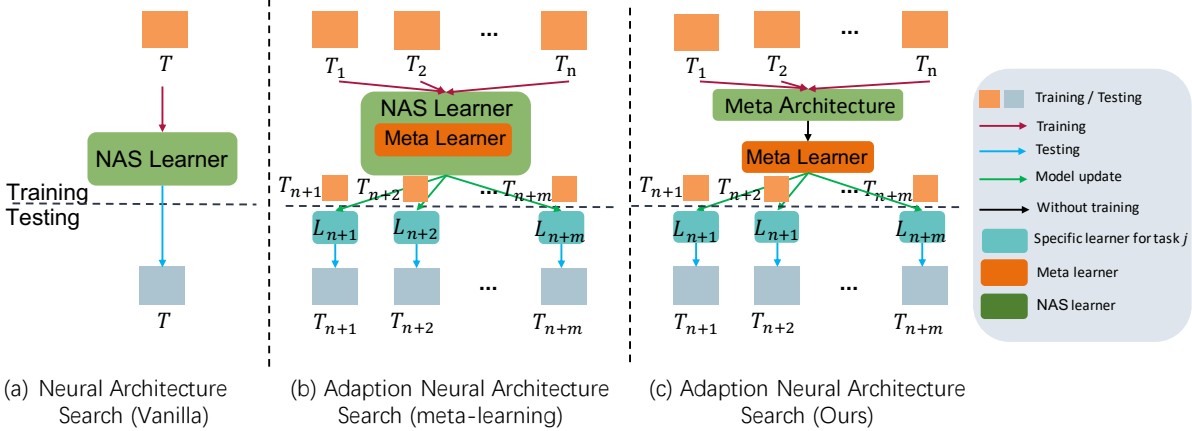

*Figure 2.* Illustration of our IBFS and related approaches. (a) Vanilla neural architecture search. (b) Adaption neural architecture search. (c) The proposed IBFS can find the best meta architecture without training for multiple unseen tasks. For the scenario of our IBFS, there is no need to restart the training of NAS after updating the meta weights for new task learning.

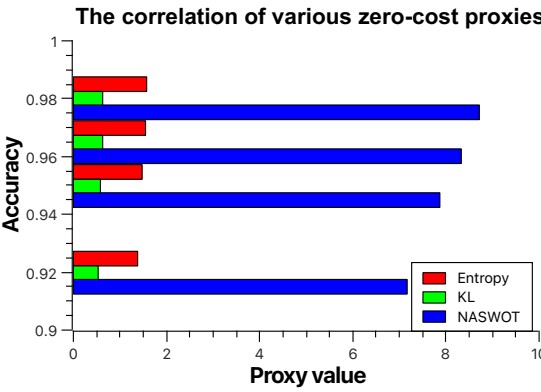

*Figure 3.* Zero-cost proxies *v.s.* Accuracy. To be specific, we sample models (e.g., densenet40, seresnet20, resnet56, pyramidnet110) from PytorchCV, as well as retrieve their real test accuracy. Then, we use various zero-cost proxies (i.e., Entropy, KL, NASWOT) to measure the expressivity of those models. For the convenience of showing, we reduced the proxy value and accuracy by 100 and 10 times, respectively. We can clearly observe that existing proxies (i.e., NWOT) suffer from larger score variance. In addition, we provide more discussion in Section 1.

costs, accuracy, and generalization than peer competitors.

**Challenges.** However, to address the limitations of the aforementioned, we target the following **key questions**:

- Which **properties** of MAML impact the global convergence of FSL?

- Can we find a **simple proxy** that well evaluates the **expressivity** of the architecture sampled from the search space for the FSL?

**Motivations.** To this end, we analyze which properties of MAML determine the direction of optimization for FSL. By carefully analyzing the global convergence of MAML, we derive Theorem 4.1, which indicates that the global convergence of MAML can be guaranteed by only considering the first-order loss landscape. Hence, Theorem 4.1 provides a shortcut to the global convergence of MAML that only considers the properties of the architecture sampled from the search space for the FSL at any initialization. Now, the problem of crafting a specialized architecture for FSL can be transferred to finding a suitable proxy to accurately measure the expressivity of architecture without training. Then, we explore the relationship between existing proxies (i.e., NASWOT) and the accuracy of architectures (as shown in Fig. 3) in a first-order way. To enhance the effectiveness of the aforementioned empirical validation, we propose two simple proxies, i.e., Kullback-Leibler (KL), and Entropy. To be specific, Fig. 3, can yield two key conclusions: ❶ No matter whether the networks at initialization or well-trained, we can clearly observe that there is a positive correlation between zero-cost proxies (Mellor et al., 2021; Chen et al., 2021b) and the actual test accuracy. In brief, it is possible to accurately measure the expressivity of architecture sampled from the search space for FSL without training. ❷ We also observe that existing proxies (i.e., NASWOT) suffer from larger score variance, which will degrade the accuracy. This is because that a larger score variance is unstable. In contrast, our proposed simple proxy, namely, Entropy, consistently obtains higher accuracy under various architectures, due to smaller score variances. Consequently, these observations suggest that designed proxies toward FSL should obey the principle of smaller score variances.

Motivated by the principle of IB theory (Tishby et al., 2000; Tishby & Zaslavsky, 2015; Shwartz-Ziv & Tishby, 2017b;

Kawaguchi et al., 2023; Hu et al., 2024), we find that the information entropy varies on different architectures, however, Kendall's Tau between accuracy and information entropy remains unchanged (as shown in Fig. 4) with the increasing epochs. Notably, the variance of the information entropy is very small, which shows that information entropy based on the IB theory is a promising proxy tailored for FSL. Grounded in empirical observations, in this paper, we propose a novel method, called IBFS, measuring the expressivity of neural architectures sampled from the predefined search space for FSL by exploring their *information entropy* properties. The *information entropy* eliminates the influence of various gradients of architectures from search spaces, making the IBFS pay more attention to the learning of multiple tasks.

**Contributions.** The key contributions of this paper are:

- To solve the bilevel optimization of MAML, we derive that the global convergence of MAML can be guaranteed by only considering the first-order approximation of the loss landscape, which transfers the problem of crafting a specialized architecture for FSL to find a suitable proxy, accurately measuring the expressivity of architectures without training.

- We propose a novel IB driven method, called Few-shot Neural Architecture Search (IBFS), which can learn a meta-architecture tailored for FSL without training.

- Extensive experiments on NAS-Bench-201 (Dong & Yang, 2020) and few-shot tasks show that IBFS achieves comparable performance in FSL with surprisingly smallest search costs, which demonstrates the effectiveness and superiority of our IBFS compared with its peer competitors.

## 2. Related Works

### 2.1. Neural Architecture Search

Recently, various Neural Architecture Search (NAS) approaches have emerged to jointly optimize the weights and the architectures (Wu et al., 2019; Chu et al., 2020b; Chen et al., 2021c; Ye et al., 2022). This is achieved by various search algorithms including Reinforcement learning (Zoph & Le, 2017; Baker et al., 2016), Evolutionary algorithm (Xie et al., 2018; Real et al., 2019b;a), and Gradient descent (Liu et al., 2019a; Yu et al., 2019; Wang et al., 2020; Chu et al., 2020b; Ye et al., 2022). To accelerate NAS, a lot of works focus on a training-free strategy (i.e., without training in the search stage), which significantly reduces search costs (Mellor et al., 2021; Xu et al., 2021; Chen et al., 2021b; Abdelfattah et al., 2021). Recently, there have been a few works to explore the generalization of architectures sampled from search spaces by combining NAS with FSL

using meta-learning (Kim et al., 2018; Lian et al., 2020; Elsken et al., 2020). However, these approaches cost more than 100 GPU days in the search stage, which is extremely time-consuming. By contrast, our proposed IBFS can find FSL-friendly architecture for new tasks without involving any training, which is very efficient. The superiority of IBFS can be attributed to the powerful IB theory, measuring the expressivity of neural architecture.

### 2.2. Information Bottleneck Theory

The IB theory (Tishby et al., 2000; Tishby & Zaslavsky, 2015; Shwartz-Ziv & Tishby, 2017b; Kawaguchi et al., 2023; Hu et al., 2024) offers a comprehensive framework for understanding machine learning models that involve latent variables (Shamir et al., 2010; Shwartz-Ziv & Tishby, 2017a; Slonim & Tishby, 1999; Tishby et al., 2000). According to this theory, in the context of input $x$, true label $y$, and latent representation $z$, the objective of supervised learning is to minimize the mutual information $I(z; x)$ while preserving as much mutual information $I(z; y)$ as possible. By achieving this, a predictor with such a representation eliminates unnecessary transformations present in $x$ while retaining crucial information about $y$. In the analysis of deep neural networks, the latent variables $z$ correspond to the hidden layer activities $z(l)$ at each layer $l$. The learning process can be visualized as a trajectory in the $(I(z; x), I(z; y))$-coordinate space. This approach, known as information plane analysis, has received significant attention in recent years (Yu et al., 2020).

### 2.3. Meta Learning for FSL

Meta-learning family (e.g., MAML (Finn et al., 2017) and its variants (Finn et al., 2019; Xu et al., 2020; Bai et al., 2021; Fallah et al., 2020b; Ji et al., 2020; Rajeswaran et al., 2019; Zhou et al., 2019; Tack et al., 2022; Sun & Gao, 2024; Kang et al., 2023)) is the most promising approach for FSL, which can quickly generalize to a new task from only a few samples by training a meta learner. Despite promising, those approaches rely on neural networks/architectures (i.e., ConvNets, ResNets) tailored for supervised learning benchmarks (i.e., ImageNet), and little attention is focused on analyzing the role of architectures for FSL. In this paper, we provide an affirmative answer by proposing a simple yet effective method, called IBFS, to design FSL-friendly architectures from a zero-cost NAS perspective. Notably, our method does not limit the scope of MAML, it can also be generalized to its variants.

## 3. Preliminaries

In this section, we introduce the concept and problem formulation of the NAS and FSL.

**Neural Architecture Search.** We consider a NAS searching process. Typically, the whole searching pipeline is organized as a Directed Acyclic Graph (DAG), suppose that we have a task $T^t$, and each task is composed of $b^t$ data samples $X^t$. Accordingly, the objective function is defined as:

$$minL_{valX^t \in T^t}(W^*, \alpha),$$
$$s.t. \quad \mathbf{W}^* = \arg\min_{\mathbf{W}} \lim (\mathbf{W}, \alpha), \quad (1)$$

where $W$ denotes the weight of the architecture from search spaces, $\alpha$ denotes the mixing weight of the operations, and $W^*$ denotes the weight of the best architecture.

**Few Shot Learning.** Suppose that we have a set of tasks $T^1$, $T^2$,..., $T^m$ (i.e., multiple datasets, multiple tasks, or multiple IoT devices, etc.). Then, each task $T^t$ is composed of $b^t$ data samples $(X^t, Y^t)$, and $(\hat{X}^t, \hat{Y}^t)$ represented by *support* samples (i.e., training samples of $D_t^{train}$) and *query* samples (i.e., test samples of $D_t^{test}$), respectively. The loss function $L$ can be formulated as follows:

$$\min_{A, W^t} \sum_t \mathcal{L}(D_t^{test}, G(D_t^{train}, \alpha; A)), t \in (1, m), \quad (2)$$

where $A$ and $W^t$ are the neural architecture and its parameters, respectively. $G$ denotes the computation of parameter updates using one or more gradient descent steps. For convenience, we employ the square loss function $\ell(\hat{Y}^t, Y^t) = \frac{1}{2}|\hat{Y}^t - Y^t|_2^2$ in this paper. As the gradient-based meta-learning algorithm, we utilize Reptile (Nichol et al., 2018), which has demonstrated comparable performance to MAML (Finn et al., 2017) in few-shot classification tasks by utilizing only first-order gradient information. However, traditional NAS methods that involve gradient-based or evolutionary computation for updates tend to have lengthy search processes. Combining such methods directly with FSL often leads to sub-optimal results.

## 4. Design of FSL-Friendly Architectures

The metrics designed for supervised learning may not be suitable for FSL, due to their optimization objective and scenario being quite different. In this section, our goal is to build a framework that yields the meta architecture $\alpha_A$ with corresponding weights $W_A$. Then, given a new task $T_i$, both $\alpha_A$ and $W_A$ should quickly adapt to $T_i$ based on a few labeled samples. To fulfill this goal, we propose IBFS, a novel method that naturally combines FSL and Zero-shot NAS. First, we analyze the convergence of MAML for the FSL. Then, we propose IBFS to evaluate the expressiveness of the architecture. In addition, we will describe how the meta architecture encoded by $\alpha_A$ can be quickly specialized to a new task without requiring re-training of $W_A$.

MAML (Model-Agnostic Meta-Learning) is a classic meta-learning approach for FSL. However, the Hessian term

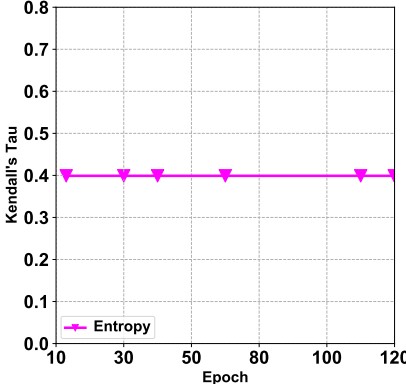

*Figure 4.* Correlation *v.s.* Epochs. The correlation coefficients (Kendall's $\tau$) between zero-cost proxy (namely, entropy) under varying number of epochs and real test accuracy of models, where those models (e.g., densenet40, seresnet20, resnet56, pyramidnet110, wrn16) are trained on CIFAR-10 datasets.

(second-order gradient) in the first-order Jacobian matrix complicates the optimization problem in MAML. If it is possible to quantify the impact of the Hessian term on the overall optimization within a certain range, it would simplify the analysis of MAML's convergence properties. Fortunately, several lines (Antoniou et al., 2018; Fallah et al., 2020a; Wang et al., 2022b; Ji et al., 2022) demonstrate that MAML can reach a global minimum that gradient descent with a sufficiently small learning rate. In this paper, we provide a proof for the global convergence of MAML as follows.

**Theorem 4.1** (*Global Convergence of MAML*). *To clearly illustrate the global convergence of MAML, we borrow one of the conclusions from MetaNTK-NAS (Wang et al., 2022b), which provides a Neural Tangent Kernel (NTK) perspective to understand MAML. While MetaNTK-NAS provides valuable theoretical guarantees, its proof is highly intricate and relies heavily on NTK theory, which can be computationally expensive. In contrast, this paper presents a novel and more accessible proof for the global convergence of MAML. Moreover, the detailed proof is provided in **App. A**. At first, we define $F_t$ sampled from search space $A$, $l$ denotes the number of $i$-th hidden layer. $\sigma_{min}(\Phi)$: The minimum singular value of a matrix $\Phi$, likely related to the Jacobian of the network's output with respect to its parameters. $\eta_0$ is a constant related to the learning. rate.$\ell_{inner} = \nabla_\theta F_t(\hat{X}^t, X^t, Y^t)$, $lr_\infty = \lim_{l \to \infty} \frac{1}{l}\ell_{inner}\ell_{inner}^T$, and $\phi_0 = \frac{2}{\xi_{\max}(\ell_\infty)+\xi_{\min}(\ell_\infty)}$. For arbitrarily small $\delta > 0$, and there exist $G > 0$, $l^* \in \mathbb{N}$,, and $\lambda_0 > 0$, hence, for width $l \geq l^*$, perform gradient descent with learning rates $lr < \frac{lr_\infty}{l}$, and $\lambda < \frac{\lambda_0}{l}$ with random initialization, from the Eq.2, we give an upper bound on the above loss function:*

$$\ell(W^t) \leq (1 - \tau \cdot \eta_0 \sigma_{min}(\Phi))^{2t}, \quad \tau \in (0, 1) \quad (3)$$

**Remark.** Theorem 4.1 indicates that there exists a global minimum with any initialization parameter $W^t$ in a small region, and the Jacobian of $\ell_{inner}$ is stable during the updating of parameter $W^t$ in the episode meta-train and meta-test. According to the theory of flatness of local minima (Keskar et al., 2016; Hoffer et al., 2017; Cha et al., 2020), the flatness of the loss landscape near the global minimum represents MAML having strong generalizability. More importantly, we can drive that the MAML is a first-order $\ell_{inner}$ convergence problem from Theorem 4.1. Then, according to the principle of IB, we evaluate the **expressivity** of the architecture sampled from the search space for the FSL by calculating the information entropy of the distribution of its inner loss $\ell_{inner}$ outputs. Fig. 4 illustrates that Kendall's Tau values for model accuracy and the expressivity of the model computed by our method *remain unchanged* across different epochs. Specifically, we sampled five models (i.e., densenet40, seresnet20, resnet56, pyramidnet110, wrn16) trained with various epochs, and calculated Kendall values using our metric in that epoch. In Fig. 4, entropy represents the score derived from our metric. Each line in the plot with six points indicates that the value of Kendall's Tau between model accuracy and entropy is the same at different epochs for multiple models. In essence, model expressivity, an intrinsic characteristic, remains stable and can be accurately computed without any training, even at initialization. This experiment validates that our proposed metric can accurately measure model expressivity without the need for training.

## 4.1. Learning Rapid Adaption of Neural Architecture via the IB Principle

**Design Principles.** Given multiple neural architectures $F_1, F_2, ..., F_n$ sampled from search space $A$ with parameter $w_i$ at initialization, when randomly sampled batch data $X$ as input, how to correctly rank these neural architectures without training becomes a key challenge. To this end, we bring the IB theoretic view and explain how it helps with ranking neural architectures at initialization. We begin by defining several terms in our analysis: $x \in X$, the output $y \in Y$, the features of neural architectures $r \in R$, and some functions for measuring information of $F_i$, i.e., entropy $H(X) = \mathbb{E}_{p(\boldsymbol{x})}[-\log p(\boldsymbol{x})]$, and Kullback-Leibler (KL) divergence $\text{KL}(p(\boldsymbol{x})\|q(\boldsymbol{x})) = \mathbb{E}_{p(\boldsymbol{x})}[\log \frac{p(\boldsymbol{x})}{q(\boldsymbol{x})}]$.

We define relevant information in a signal $x \in X$ as that signal providing information about another signal $y \in Y$.

Examples include information implicit in image tasks or neural architectures. The main inquiry is to explore those features of $x$ that play a role in the prediction of the neural architecture's output $y$. We define this problem as squeezing the information about $x$ provided by $y$ through a "bottleneck" formed by some metric on the neural architecture. Given a neural architecture $F_i$ at initialization, $Y = F_i(X)$, we try to construct a function for measuring the expressivity of a neural architecture based on the following IB principle. As follows, the amount of information about $Y$ in $R$ is given by:

$$I(R; Y) = \sum_y \sum_r p(y, r) \log \frac{p(y, r)}{p(y)p(r)} \leq I(X; Y), \quad (4)$$

According to (Tishby et al., 2000), we give an objective function that finds the trade-off between the input $x$ and neural architecture $F_i$ output $y$, the objective function can be formulated as follows:

$$\mathcal{L}[p(r|x)] = I(R; X) - \beta I(R; Y), \quad (5)$$

where $\beta \geq 0$ is the Lagrange multiplier attached to the constrained meaningful information, while maintaining the normalization of the mapping $p(r|x)$ for every $x$. Introducing $\lambda(x)$ for the normalization of the conditional distributions $p(r|x)$, the Eq.5, becomes:

$$\mathcal{L}[p(r|x)] = I(X; R) - \beta I(R; Y) - \sum_{x,r} \lambda(x)p(r|x)$$

$$= \sum_x \sum_r p(r, x) \log \frac{p(r, x)}{p(x)p(r)}$$

$$- \sum_y \sum_r p(y, r) \log \frac{p(y, r)}{p(y)p(r)} - \sum_{x,r} \lambda(x)p(r|x) \quad (6)$$

$$\leq \sum_x \sum_r p(r, x) \log \frac{p(r, x)}{p(x)p(r)}$$

$$- \beta I(X; Y) - \sum_{x,r} \lambda(x)p(r|x),$$

Finally, we obtain the final optimal solution by minimizing Eq.6:

$$\frac{\delta \mathcal{L}}{\delta p(r|x)} =$$

$$p(x) \left[ \log \frac{p(r|x)}{p(r)} + \beta \sum_y p(y|x) \log \frac{p(y|x)}{p(y|r)} - \tilde{\lambda}(x) \right] = 0,$$

$$(7)$$

where $\tilde{\lambda}(x) = \frac{\lambda(x)}{p(x)} - \beta \sum_y p(y|x) \log \left[ \frac{p(y|x)}{p(y)} \right]$. If we choose entropy $H(X) = \mathbb{E}_{p(\boldsymbol{x})}[-\log p(\boldsymbol{x})]$ as the function to measure distortion of neural architecture $F_i$, the $p(r|x)$, becomes:

*Table 1.* Comparison results on NAS-Bench-201. Red, blue, and orange indicate the best, second-best, and third-best results, respectively.

| Method | Year | Search (s) | CIFAR-10 | | CIFAR-100 | | ImageNet-16-120 | | Search Methods |
|---|---|---|---|---|---|---|---|---|---|
| | | | validation (%) | test (%) | validation (%) | test (%) | validation (%) | test (%) | |
| ResNet(He et al., 2016) | CVPR2016 | | 93.97 | | | 70.86 | | 43.63 | Manual |
| | | | **Non-weight sharing** | | | | | | |
| REA(Zoph et al., 2018) | CVPR2018 | 12000 | 91.19±0.31 | 93.92±0.30 | 71.81±1.12 | 71.84±0.99 | 45.15±0.89 | 45.54±1.03 | EA |
| BOHB(Liu et al., 2018) | ECCV2018 | 12000 | 90.82±0.53 | 93.61±0.52 | 70.74±1.29 | 70.85±1.28 | 44.26±1.36 | 44.42±1.49 | HPO |
| REINFORCE(Real et al., 2019b) | AAAI2019 | 12000 | 91.09±0.37 | 93.85±0.37 | 71.61±1.12 | 71.71±1.09 | 45.05±1.02 | 45.24±1.18 | RL |
| | | | **Weight sharing** | | | | | | |
| SNAS (Xie et al., 2020) | ICLR2018 | - | 90.10±1.04 | 92.77±0.83 | 69.69±2.39 | 69.34±1.98 | 42.84±1.79 | 43.16±2.64 | GD |
| ENAS(Pham et al., 2018) | ICML2018 | 13315 | 39.77±0.00 | 54.30±0.00 | 15.03±0.00 | 15.61±0.00 | 16.43±0.00 | 16.32±0.00 | RL |
| DARTS-V2(Liu et al., 2019b) | ICLR2019 | 29902 | 39.77±0.00 | 54.30±0.00 | 15.03±0.00 | 15.61±0.00 | 16.43±0.00 | 16.32±0.00 | GD |
| GDAS(Dong & Yang, 2019) | CVPR2019 | 28926 | 90.00±0.21 | 93.51±0.13 | 71.14±0.27 | 70.61±0.26 | 41.70±1.26 | 41.84±0.90 | GD |
| DSNAS (Hu et al., 2020) | ICLR2019 | - | 89.66±0.29 | 93.08±0.13 | 30.87±16.40 | 31.01±16.38 | 40.61±0.09 | 41.07±0.09 | GD |
| PC-DARTS (Xu et al., 2019) | ICLR2020 | - | 89.96±0.15 | 93.41±0.30 | 67.12±0.39 | 67.48±0.89 | 40.83±0.08 | 41.31±0.22 | GD |
| RSPS(Li & Talwalkar, 2020) | UAI2020 | 7587 | 84.16±1.69 | 87.66±1.69 | 59.00±4.60 | 58.33±4.34 | 31.56±3.28 | 31.14±3.88 | RS+WS |
| iDARTS (Zhang et al., 2021a) | ICML2021 | - | 89.86±0.60 | 93.58±0.32 | 70.57±0.24 | 70.83±0.48 | 40.38±0.59 | 40.89±0.68 | GD |
| OLES (Jiang et al., 2023) | NeurIPS2023 | - | 90.88±0.10 | 93.70±0.15 | 70.56±0.28 | 70.40±0.22 | 44.17±0.49 | 43.97±0.38 | GD |
| IS-DARTS (He et al., 2024) | AAAI2024 | 7200 | 91.55±0.00 | 94.36±0.00 | 73.49±0.00 | 73.31±0.00 | 46.37±0.00 | 46.34±0.00 | GD |
| | | | **Training-free** | | | | | | |
| Random | | - | 83.20 ± 13.28 | 86.61 ± 13.46 | 60.70 ± 12.55 | 60.83 ± 12.58 | 33.34 ± 9.39 | 33.13 ± 9.66 | Random |
| NASWOT (Mellor et al., 2021) | ICML2021 | 4.4 | 88.47 ± 1.33 | 91.53 ± 1.62 | 66.49 ± 3.08 | 66.63 ± 3.14 | 38.33 ± 4.98 | 38.33 ± 5.22 | Zero-cost |
| GradSign (Zhang & Jia, 2022) | ICLR2022 | 30.38 | - | 93.52 ± 0.19 | - | 70.57 ± 0.31 | - | 41.89 ± 0.69 | Zero-cost |
| ZiCo (Li et al., 2023) | ICLR2023 | 6.2 | - | 93.50 ± 0.18 | - | 70.62 ± 0.26 | - | 42.04 ± 0.82 | zero-cost |
| AZ-NAS (Lee & Ham, 2024) | CVPR2024 | 0.71 | - | 93.53 ± 0.15 0.723 | - | 70.75 ± 0.48 | - | 45.43 ± 0.29 | zero-cost |
| SWAP (Peng et al., 2024) | ICLR2024 | 4.7 | 87.31 ± 2.36 | 90.48 ± 0.94 | 65.92 ± 4.32 | 67.13 ± 1.83 | 33.85 ± 4.98 | 35.40 ± 3.96 | Zero-cost |
| **IBFS(ours)** | | 3.82 | 91.55 ± 0.76 | 94.37 ± 0.34 | 73.31 ± 2.12 | 73.09 ± 2.08 | 45.59 ± 0.32 | 46.33 ± 1.2 | Zero-cost |
| Optimal (NAS-Bench-201) | | N/A | 94.37 | | 73.51 | | 47.31 | | N/A |

$$p(r|x) = \frac{p(\tilde{r})}{\tilde{\lambda}(x)} exp(-\beta H(X))$$

$$= \frac{p(\tilde{r})}{\tilde{\lambda}(x)} exp(-\beta \mathbb{E}_{p(\boldsymbol{x})}[-\log p(\boldsymbol{x})]) \quad (8)$$

$$\leq H * exp(-\beta \mathbb{E}_{p(\boldsymbol{x})}[-\log p(\boldsymbol{x})]),$$

where $H$ is a scalar.

Given a neural architecture $F_i$ at initialization, first, we measure the expressivity of a neural architecture based on the above IB principle. Concretely, we define a mapping from $X$, through untrained architecture $F_i$ with parameter $w_i$, construction of kernel Jacobian matrix, input a batch with $B$ data $x_j$, where $x_j \in X$, $j \leq B$, $J_i = \frac{\partial w(x_i)}{\partial x_i}$. To evaluate the performance of the architecture $F_i$ on different inputs data $x_j$, we constructed the following Jacobian matrix $J$:

$$J = \left( \frac{\partial w(x_1)}{\partial x_1} \cdots, \frac{\partial w(x_B)}{\partial x_B} \right)^T, \quad (9)$$

The intuition to our method is that different neural architectures have different Jacobian $J$ values, based on the $N$ eigenvalues $\varsigma_1, \varsigma_2, ..., \varsigma_N$ of the Jacobian matrix, we construct an entropy-based IB formulation for the neural architecture $F_i$. The untrained architecture $F_i$ can be scored as:

$$NN_{expressivity} = -\sum_{k=1}^{N} p \log p \quad (10)$$

$$\leq H * exp(-\mathbb{E}_{p(\varsigma_{\boldsymbol{k}})}[-\log p(\varsigma_{\boldsymbol{k}})]).$$

The intuition for entropy-driven IB is that a higher score

at initialization implies the network architecture obtains a higher accuracy after convergence.

## 5. Experiments

### 5.1. Experiment Setup

To validate the effectiveness of the proposed proxy, we first evaluate our IBFS framework on search space of NAS-Bench-201(Dong & Yang, 2020) in three supported datasets (CIFAR-10, CIFAR-100, ImageNet-16-120 (Chrabaszcz et al., 2017)). Then, to validate the effectiveness of our IBFS in designing the FSL-friendly architecture, we conduct comprehensive experiments in two popular few-shot image classification datasets, mini-ImageNet, and tiered-ImageNet with the wide of peer competitors, which both are subsets of ImageNet (Deng et al., 2009). (1) *mini-ImageNet* (Vinyals et al., 2016): It contains 60,000 RGB images of 84x84 pixels extracted from ImageNet1K (Deng et al., 2009). It includes 100 classes (each with 600 images) that are split into 64 training classes, 16 validation classes, and 20 test classes. (2) *tiered-ImageNet* (Ren et al., 2018): This dataset contains 779,165 RGB images of 84x84 pixels extracted from ImageNet1K (Deng et al., 2009). It includes 608 classes that are split into 351 training, 97 validation, and 160 test classes.

**Peer Competitors.** For FSL, we compare our method with a wide scope of the state-of-the-art NAS-based baselines, as follows: (1) MAML(Finn et al., 2017); (2) ANIL (Raghu et al., 2020); (3) MetaOptNet(Lee et al., 2019); (4) RFS(Tian et al., 2020); (5) AutoMeta(Kim et al., 2018); (6) T-NAS++(Lian et al., 2020); (7) MetaNAS(Elsken et al., 2020); (8) H-Meta-NAS(Zhao et al., 2022); (9) MetaNTK-

*Table 2.* Performance comparison on the DARTS search space with ImageNet1k dataset, where "Img" denotes network models directly searched in ImageNet1k, and "C10" and "C100" denotes network models searched in CIFAR-10 and CIFAR-100, respectively.

| Method | Top-1(%) | Top-5(%) | # Params (M) | FLOPS (M) | Search Cost (GPU-days) | Search Method |
|---|---|---|---|---|---|---|
| ResNet50 (He et al., 2016) | 75.3 | 92.2 | 25.6 | 4100 | - | Manual |
| MobileNetV1 (Howard et al., 2017) | 70.6 | 89.5 | 4.2 | 575 | - | Manual |
| MobileNetV2 (Sandler et al., 2018) | 74.7 | 91.0 | 3.4 | 300 | - | Manual |
| ShuffleNetV2 (Ma et al., 2018) | 72.6 | - | 3.5 | 299 | - | Manual |
| AmoebaNet-A (Zoph et al., 2018) | 74.5 | 92.4 | 6.4 | 555 | 3150 | Evolution |
| ProxylessNAS-RL (Cai et al., 2018) | 74.6 | 92.3 | 5.8 | 465 | 8.3 | RL |
| EfficientNet-B0 (Tan & Le, 2019) | 76.3 | 93.2 | 5.3 | 390 | ≈3000 | RL |
| NASNet-A (Real et al., 2019b) | 74.0 | 91.6 | 5.3 | 564 | 2000 | RL |
| DARTS (Liu et al., 2019b) | 73.3 | 91.3 | 4.7 | 574 | 4 | Gradient |
| FBNet (Wu et al., 2019) | 74.9 | - | 5.5 | 375 | 216 | Gradient |
| PC-DARTS(Img) (Xu et al., 2019) | 75.8 | 92.7 | 5.3 | 597 | 3.7 | Gradient |
| P-DARTS(C100) (Chen et al., 2019) | 75.3 | 92.5 | 5.1 | 577 | 0.3 | Gradient |
| DARTS+ (Liang et al., 2019) | 76.3 | 92.8 | 5.1 | 591 | 0.2 | Gradient |
| DARTS-(Img) (Chu et al., 2020a) | 76.2 | 93.0 | 4.9 | 467 | 4.5 | Gradient |
| FairDARTS-B(Img) (Chu et al., 2020b) | 75.1 | 92.5 | 4.8 | 541 | - | Gradient |
| SNAS(C10) (Xie et al., 2020) | 72.7 | 90.8 | 4.3 | 522 | 1.5 | Gradient |
| DARTS+PT(C10) (Wang et al., 2020) | 74.5 | 92.0 | 4.6 | - | 0.8 | Gradient |
| DOTS(C10) (Gu et al., 2021) | 75.7 | 92.6 | 5.2 | 581 | 0.3 | Gradient |
| β-DARTS(C100) (Ye et al., 2022) | 75.8 | 92.9 | 5.4 | 597 | 0.4 | Gradient |
| Λ-DARTS (Movahedi et al., 2023) | 75.7 | - | 5.2 | - | - | Gradient |
| OLES (Jiang et al., 2023) | 75.5 | 92.6 | 4.7 | - | 0.4 | Gradient |
| FP-DARTS(C10) (Wang et al., 2023) | 75.7 | 92.7 | 5.4 | - | 0.08 | Gradient |
| PDARTS-$AER^{ad}$ (Jing et al., 2023) | 76.0 | 92.8 | 5.1 | 578 | 2.0 | Gradient |
| IS-DARTS (He et al., 2024) | 75.9 | 92.9 | 6.4 | - | 0.42 | Gradient |
| NAO (Luo et al., 2018) | 74.3 | 91.8 | 11.4 | 584 | 200 | Proxy |
| SemiNAS (Luo et al., 2020) | 76.5 | 93.2 | 6.3 | 599 | 4 | Proxy |
| WeakNAS (Wu et al., 2021) | 76.5 | 93.2 | 5.5 | 591 | 2.5 | Proxy |
| TENAS (Chen et al., 2021c) | 75.5 | 92.5 | 5.4 | - | 0.17 | Training-free |
| NASI-ADA(C10) (Shu et al., 2022) | 75.0 | 92.2 | 4.9 | 559 | 0.01 | Training-free |
| SWAP (Shu et al., 2022) (Img) | 75.0 | 92.4 | 5.8 | - | 0.006 | Training-free |
| **IBFS**(C10) | 76.7 ↑(0.2) | 93.5 ↑(0.3) | 5.2 | 587 | 0.0042 ↓(0.0022) | training-free |

NAS(Wang et al., 2022a). In addition, we also compare our method with a wide scope of the state-of-the-art non-NAS-based baselines, as follows: (1) MAML(Finn et al., 2017); (2) ANIL (Raghu et al., 2020); (3) COMLN(Deleu et al., 2022); (4) Meta-AdaM(Sun & Gao, 2024); (5) GAP(Kang et al., 2023); (6) MetaDiff(Zhang et al., 2024); (7) MetaOptNet(Lee et al., 2019); (8) CTM(Li et al., 2019); (9) RFS(Tian et al., 2020); (10) MAML+ALFA(Baik et al., 2020); (11) Sparse-MAML(Von Oswald et al., 2021); (12) MeTAL(Baik et al., 2021); (13) ClassifierBaseline(Chen et al., 2021d); (14) MetaQDA(Zhang et al., 2021b); (15) MAML+SiMT(Tack et al., 2022); (16) COMLN(Deleu et al., 2022).

**Implementation Details.** In our study, we construct the neural architecture by stacking either 5 or 8 searched cells together. Within the architecture, there are specific cells positioned at 1/3 and 2/3 of the total depth of the network, referred to as reduction cells. These reduction cells serve the purpose of decreasing the spatial resolution while doubling the number of channels. The initial number of channels is set to 48 as a starting point for the architecture.

**Optimization Setup.** In line with the approach proposed in (Tian et al., 2020), we employ the stochastic gradient descent (SGD) optimizer with a momentum of 0.9 and a weight decay of 0.0005. The training process for all models consists of 120 epochs for miniImageNet and 80 epochs for tieredImagenet. Regarding the specific learning rate schedules, for miniImageNet, we start with an initial learning rate of 0.1. At epochs 40 and 80, the learning rate is decayed by

a factor of 10x. As for tieredImageNet, the initial learning rate is set to 0.2. We apply a 10x learning rate decay at epochs 20, 40, and 60, 80.

**Metrics.** In line with common practice, we give priority to the relative ranking of architectures and utilize Kendall's Tau (Sen, 1968) and accuracy as the evaluation metrics. Kendall's Tau is a widely adopted measure for assessing the quality of performance predictors. It aims to evaluate the relative ranking of candidate architectures and provides values within the range of [-1, 1]. A higher Kendall's Tau indicates a stronger alignment between the predicted ranking and the actual ranking, thereby indicating better performance.

**Code.** We have implemented our code using the PyTorch framework (Paszke et al., 2019). Specifically, for the NAS search stage, we have built upon the codebase provided by (Chen et al., 2021b). This serves as the foundation for our implementation. On the other hand, for the training and evaluation stages, we have utilized the code provided by (Tian et al., 2020).

**Hardware.** The majority of our experiments were conducted using NVIDIA RTX 2080Ti GPUs, while the remaining experiments were run on NVIDIA RTX A100 80G GPUs. Each experiment was executed on a single GPU at a time to ensure consistent and reliable results. The search cost of IBFS, specifically, was benchmarked using NVIDIA RTX 2080Ti GPUs.

*Table 3.* Comparison results on FSL benchmarks. Red, blue, and orange indicate the best, second-best, and third-best results, respectively.

| Method | Year | Arch. | #Cells | Train | mini-ImageNet 5-way | | | | tiered-ImageNet 5-way | | | |
| --- | --- | --- | --- | --- | --- | --- | --- | --- | --- | --- | --- | --- |
| | | | | | #Param. | Search Cost | 1-shot (%) | 5-shot (%) | #Param. | Search Cost | 1-shot (%) | 5-shot (%) |
| MAML(Finn et al., 2017) | ICML17 | Conv4 | - | MAML | 30k | - | 48.70±1.84 | 63.11±0.92 | 30k | - | 51.67±1.81 | 70.30±1.75 |
| ANIL (Raghu et al., 2020) | ICLR20 | Conv4 | - | ANIL | 30k | - | 48.0±0.7 | 62.2±0.5 | - | - | - | - |
| COMLN(Deleu et al., 2022) | LCLR22 | Conv4 | - | - | - | - | 53.01±0.62 | 70.54±0.54 | - | - | 54.30±0.69 | 71.35±0.57 |
| Meta-AdaM(Sun & Gao, 2024) | NeurIPS23 | Conv4 | - | Meta Learning | - | - | 52.00±0.49 | 70.70±0.49 | - | - | 53.93±0.49 | 72.66±0.49 |
| GAP(Kang et al., 2023) | CVPR23 | Conv4 | - | - | - | - | 54.86±0.85 | 71.55 ±0.61 | - | - | 57.60±0.93 | 74.90±0.68 |
| MetaDiff(Zhang et al., 2024) | AAAI24 | Conv4 | - | Meta Learning | - | - | 55.06±0.81 | 73.18 ±0.64 | - | - | 57.77±0.90 | 75.46±0.69 |
| MetaOptNet(Lee et al., 2019) | CVPR19 | ResNet-12 | - | MetaOptNet | 12.5M | - | 62.64±0.61 | 78.63±0.46 | 12.7M | - | 65.99±0.72 | 81.56±0.53 |
| CTM(Li et al., 2019) | CVPR19 | ResNet-18 | - | - | - | - | 64.12±0.82 | 80.51±0.13 | - | - | 68.41±0.39 | 84.28±1.73 |
| RFS(Tian et al., 2020) | ECCV20 | ResNet-12 | - | RFS | 12.5M | - | 62.02±0.63 | 79.64±0.44 | 12.7M | - | 69.74±0.72 | 84.41±0.55 |
| MAML+ALFA(Baik et al., 2020) | NeurIPS20 | ResNet-12 | - | MAML | - | - | 59.74±0.49 | 77.96±0.47 | - | - | 64.62±0.49 | 82.48±0.39 |
| Sparse-MAML(Von Oswald et al., 2021) | NeurIPS21 | ResNet-12 | - | MAML | - | - | 56.39±0.38 | 73.01±0.24 | - | - | 53.47±0.53 | 68.83±0.65 |
| MeTAL(Baik et al., 2021) | ICCV21 | ResNet-12 | - | Meta Learning | - | - | 59.64±0.38 | 76.20±0.19 | - | - | 63.89±0.48 | 80.14±0.40 |
| ClassifierBaseline(Chen et al., 2021d) | ICCV21 | ResNet-12 | - | Meta Learning | - | - | 61.22±0.84 | 78.72±0.60 | - | - | 69.71±0.88 | 83.87 ±0.64 |
| MetaQDA(Zhang et al., 2021b) | ICCV21 | ResNet-12 | - | Bayesian | - | - | 65.12 ±0.66 | 80.98±0.75 | - | - | 69.97 ±0.52 | 85.51 ±0.58 |
| MAML+SiMT(Tack et al., 2022) | NeurIPS22 | ResNet-12 | - | MAML | - | - | 51.49±0.18 | 68.74±0.12 | - | - | 52.51±0.21 | 69.58±0.11 |
| COMLN(Deleu et al., 2022) | LCLR22 | ResNet-12 | - | - | - | - | 59.26±0.65 | 77.26±0.49 | - | - | 62.93±0.71 | 81.13 ±0.53 |
| Meta-AdaM(Sun & Gao, 2024) | NeurIPS23 | ResNet-12 | - | Meta Learning | - | - | 59.89±0.49 | 77.92±0.43 | - | - | 65.31±0.48 | 85.24±0.35 |
| MetaDiff(Zhang et al., 2024) | AAAI24 | ResNet-12 | - | Meta Learning | - | - | 64.99±0.77 | 81.21 ±0.56 | - | - | 72.33±0.92 | 86.31±0.62 |
| AutoMeta(Kim et al., 2018) | NeurIPS18 | Cells | - | Reptile | 100k | 2688 hr | 57.6±0.2 | 74.7±0.2 | - | - | - | - |
| T-NAS++(Lian et al., 2020) | ICLR20 | Cells | 2 | FOMAML | 27k | 48 hr | 54.11±1.35 | 69.59±0.85 | - | - | - | - |
| MetaNAS(Elsken et al., 2020) | CVPR20 | Cells | 5 | Reptile | 1.1M | 168 hr | 63.1±0.3 | 79.5±0.2 | - | - | - | - |
| MetaNAS(Elsken et al., 2020)(retrained)[†] | CVPR20 | Cells | 8 | RFS | 3.53M | 168 hr | 63.88±0.23 | 79.88±0.14 | 3.70M | 168 hr | 72.32±0.02 | 86.48±0.06 |
| H-Meta-NAS(Zhao et al., 2022) | NeurIPS22 | - | - | - | 70.28K | - | 57.36±1.11 | 77.53±0.77 | - | - | - | - |
| MetaNTK-NAS(Wang et al., 2022a) | CVPR22 | Cells | 8 | RFS | 3.21M | 1.92 hr | 64.26±0.14 | 80.35±0.12 | 4.78M | 2.73 hr | 72.37±0.79 | 86.43±0.52 |
| **IBFS** | - | Cells | 8 | RFS | 3.50M | 0.1 hr | 64.55±0.02 | 81.52±0.08 | 4.50M | 0.10 hr | 72.56±0.02 | 86.73±0.08 |

## 5.2. Results on NAS-Bench-201

Table 1 details the comparison between our IBFS and its peer competitors in terms of NAS benchmark (namely, NAS-Bench-201 search space) in CIFAR-10, CIFAR-100, and ImageNet-16-120 datasets. To be specific, as depicted in Table 1, the non-weight sharing methods suffer from huge search costs, which search the target architecture with a budget of 12000 seconds. For training-free and non-weight-sharing methods, we report the accuracies are averaged over 500 runs. As shown, it is evident that the IBFS consistently outperforms its peer competitors in all cases. In particular, IBFS only cost 3.82s, finding competitive architecture at initialization than other methods. In addition, our IBFS obtains a higher similarity value in terms of Kendall's Tau than the existing zero-cost proxy (i.e., NASWOT) designed for NAS (0.752 vs. 0.422). Notably, NASWOT is designed for NAS in classification tasks, not for few-shot learning. While NASWOT and our method utilize the Jacobian matrix, however, the computation of the Jacobian matrix is totally different. NASWOT uses the Hamming distance between two binary codes. By contrast, we directly utilize the gradient of the eutwork kernel. To sum up, this experiment provides a piece of critical evidence that proposed IBFS can design FSL-friendly architectures without training.

## 6. Results in Larger ImageNet1k

To validate the effectiveness of our method on larger dataset, we conduct the experiments in ImageNet1k. From Table 2, we can clearly see that our IBFS method consistently outperforms compared SOTA methods: it achieves highest 76.7 % Top-1 accuracy, with 0.0042 (GPU-days) fewest search costs. Compared with its peer competitors, our

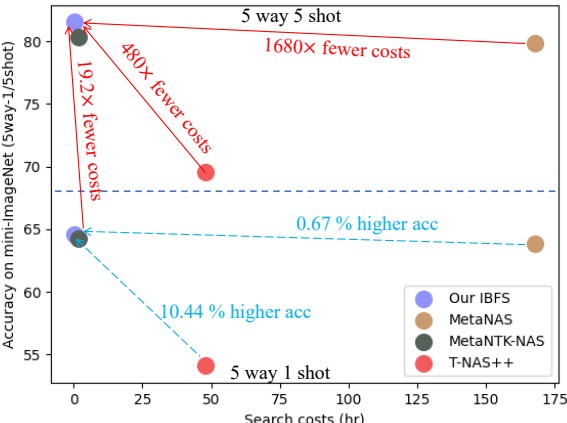

*Figure 5.* Comparison to SOTAs in terms of **accuracy of FSL** and **Search costs** in mini-ImageNet dataset (5 way 5 shot).

method achieves the Top-1 accuracy of 1.7% higher than SWAP, 1.6% higher than NASI-ADA, and 1.2% higher than TENAS. Notably, our method only searches in small CIFAR-10 dataset, and then can well generalize to large ImageNet1k, which largely reduces search costs. Those empirical results illustrate that our method possesses strong generalization ability in larger dataset.

### 6.1. Results for Few Shot Learning

Table 3 presents a comparison of IBFS with a wide scope of meta-learning approaches under the setting of 5-way few-shot classification tasks in miniImageNet and tiered-ImageNet datasets. Notably, to keep a fair comparison, we

only compare with CNN-based methods. If we use transformers as the backbone in FSL, it will bring unfairness for comparison. Among these approaches, MetaDiff represents the current state-of-the-art gradient-based meta-learning approach for FSL, which obtains $64.99\pm0.77$ (1 shot) and $81.21\pm0.56$ (5 shot) in miniImageNet dataset using ResNet-12 as a feature extractor. Additionally, we present a comparison of IBFS with a wide scope of NAS-based methods (i.e., AutoMeta, MetaNAS, and MetaNTK-NAS) for FSL. In this case, our FSL consistently surpasses state-of-the-art methods in terms of accuracy and search costs, showing superiority in terms of search speed tailored for FSL. To put it more intuitively, we present the visualization of comparison in Fig. 5. The result of Fig. 5 suggests that the architecture guided by IBFS more effectively learns features for FSL, facilitating adaption to unseen tasks. Notably, as shown Fig. 7 and 8, we provide the visualization of the searched architecture, which intuitively validates the effectiveness of our method. Overall, these results validate the effectiveness of IBFS tailored for FSL.

### 6.2. Results for Transformer Design

To further scrutinize the effectiveness of our method for transformer designing, we devote lots of effort & exploration in the IBFS day and night, conducting additional experiments on AutoFormer (Chen et al., 2021a) in a larger ImageNet dataset. The experimental setting is the same as TF-TAS-T (Zhou et al., 2022). We can find that IBFS achieves the highest Top-1 accuracy (76.5%). Those empirical results show the strong generalizability of our method for transformer design.

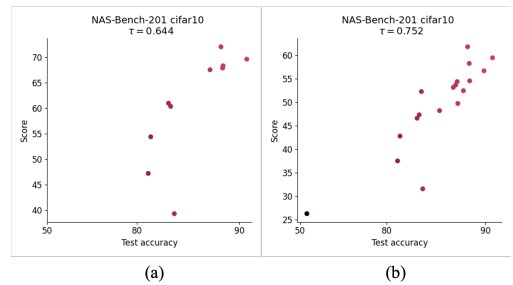

*Figure 6.* The impact of $\theta$ on NAS-Bench-201 search space. (a) and (b) are Kendall's $\tau$ on various hyper-parameters $\alpha$.

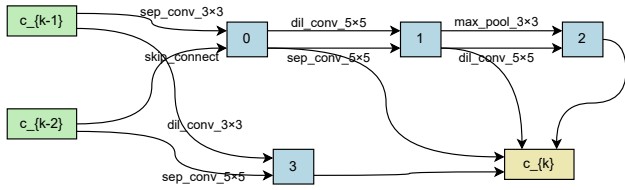

*Figure 7.* Normal cell searched by IBFS.

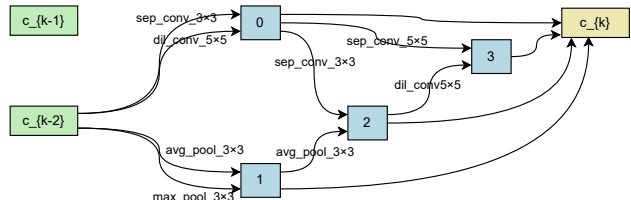

*Figure 8.* Reduce cell searched by IBFS.

*Table 4.* Results on AutoFormer benchmark in ImageNet.

| Method | Year | Top-1 (%) | Search Cost | Model Type | Search Method |
|---|---|---|---|---|---|
| ViT-Ti(Dosovitskiy et al., 2020) | ICLR2020 | 74.5 | - | Transformer | Manual |
| AutoFormer-T (Chen et al., 2021a) | CVPR2021 | 74.9 | 24 | Transformer | Evolution |
| TF-TAS-T (Zhou et al., 2022) | CVPR2022 | 75.3 | 0.5 | Transformer | Training-free |
| ViTAS-C (Su et al., 2022) | ECCV2022 | 74.7 | 32 | Transformer | Evolution |
| Auto-Prox (Wei et al., 2024) | AAAI2024 | 75.6 | 0.1 | Transformer | Training-free |
| **IBFS** | - | 76.5 | 0.03 | CNNs | Training-free |

## 7. Conclusion and Future work

This paper proposes a novel IB driven Few-shot Neural Architecture Search (IBFS) framework to address the challenge of designing high-performance neural architectures for new tasks without involving any training. We first demonstrate that the global convergence of Model-agnostic meta-learning (MAML) can be guaranteed by considering only the first-order loss landscape. Additionally, we leverage the IB theory to develop a training-free method for FSL, enabling to effectively design and rank neural architectures based on IB theory. We conduct extensive experiments to validate the effectiveness of our IBFS framework. The experimental results show that our IBFS achieves state-of-the-art performance in terms of accuracy and search costs in FSL without the need for training.

In the future, we plan to deploy IBFS on edge devices (i.e., IoT, and Raspberry Pi), which will significantly benefit the NAS community and real-world applications by providing high-quality services in terms of discovering high-performing architectures. Because edge devices possess the characteristics of limited resources, it highlights the need for more efficient and high-performing architectures by deploying our IBFS. However, it is challenging to deploy IBFS on edge devices due to the limited resources of edge devices. Therefore, we plan to explore ways to reduce the costs of IBFS in terms of storage and search costs. In addition, to enhance the generalization of our method under cross-domain settings, we will study cross-domain FSL for more complicated and challenging, varying-way, varying-shot tasks beyond the current scope.

## Impact Statement

This paper presents work whose goal is to advance the field of Machine Learning. There are many potential societal consequences of our work, none which we feel must be specifically highlighted here.

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

## Overview of the Appendix

The overview of this appendix is as follows:

- App. A: Describes detailed proof of global convergence of MAML.

## A. Proof of Global Convergence of MAML

In this section, we will prove the global convergence of MAML. First, we rethink the process of MAML under the FSL setting, then, we introduce some variables used in the proof. Finally, we will introduce proof of global convergence of MAML.

**Rethink the Process of MAML.** From the FSL definition, we have a set of tasks $\{\mathcal{T}^t\}_{t=1}^m$, where each task $\mathcal{T}^t$ consists of $b^t$ data samples. These are split into support (training) samples $(\hat{X}^t, \hat{Y}^t) \in D_t^{\text{train}}$ and query (test) samples $(X^t, Y^t) \in D_t^{\text{test}}$. The loss function for FSL is defined as:

$$\min_{A,W^t} \sum_{t=1}^m \mathcal{L}(D_t^{\text{test}}, G(D_t^{\text{train}}, \alpha; A)), \tag{11}$$

where $A$ is the neural architecture, $W^t$ are the task-specific parameters, and $G$ represents parameter updates via gradient descent. From the NAS definition, the search process is a Directed Acyclic Graph (DAG), and the objective is:

$$\min L_{\text{val}X^t \in \mathcal{T}^t}(W^*, \alpha), \quad \text{s.t.} \quad W^* = \arg\min_W L_{\text{train}X^t \in \mathcal{T}^t}(W, \alpha), \tag{12}$$

where $W$ are the weights of the architecture, $\alpha$ are the mixing weights of operations, and $W^*$ is the optimal weight. In the paper, MAML is formulated as a bi-level optimization problem:

$$\boldsymbol{\theta}^* = \arg\min_{\boldsymbol{\theta} \in \Theta} F(\boldsymbol{\theta}) := \frac{1}{M} \sum_{i=1}^M F_i(\boldsymbol{\theta}), \quad F_i(\boldsymbol{\theta}) = f_i(\phi_i(\boldsymbol{\theta})), \quad \phi_i(\boldsymbol{\theta}) = \text{Alg}(\hat{f}_i, \boldsymbol{\theta}, \boldsymbol{h}), \tag{13}$$

where $\hat{f}_i$ is the training loss, $f_i$ is the test loss, and Alg is an optimization algorithm (e.g., gradient descent) that adapts $\boldsymbol{\theta}$ to task-specific parameters $\phi_i(\boldsymbol{\theta})$.

**Definition of Variable.** In this section, we provide the detailed definition of variables in our paper. The details are as follows:

- $F_t$: This represents the loss for task $\mathcal{T}^t$, sampled from a search space $A$. Based on the FSL definition, $F_t(\hat{X}^t, X^t, Y^t)$ is the test loss $f_t(\phi_t(\boldsymbol{\theta}))$, where $\phi_t(\boldsymbol{\theta})$ is the adapted parameter for task $\mathcal{T}^t$.

- $\ell(W^t) = \ell(\hat{Y}^t, Y^t) = \frac{1}{2}\|F_t(\hat{X}^t, X^t, Y^t) - \hat{Y}^t\|_2^2$: The loss is the squared error between the predicted output $\hat{Y}^t = F_t(\hat{X}^t, X^t, Y^t)$ (using adapted parameters) and the true output $Y^t$. Here, $W^t$ corresponds to the adapted parameters $\phi_t(\boldsymbol{\theta})$.

- $\ell_{\text{inner}} = \nabla_\theta F_t(\hat{X}^t, X^t, Y^t)$: This is the gradient of the task-specific loss with respect to the meta-parameters $\boldsymbol{\theta}$, computed on the support set.

- $lr_\infty = \lim_{l \to \infty} \frac{1}{l} \ell_{\text{inner}} \ell_{\text{inner}}^T$: This appears to define a limiting behavior of the inner gradient's covariance as the width $l$ (number of hidden units in layer $i$) increases. It's likely intended to capture the spectral properties of the gradient.

- $\phi_0 = \frac{2}{\xi_{\max}(\ell_\infty) + \xi_{\min}(\ell_\infty)}$: Here, $\xi_{\max}$ and $\xi_{\min}$ are the maximum and minimum eigenvalues of $\ell_\infty$, suggesting $\phi_0$ is related to the learning rate or a stability constant.

- $\eta_0$: A constant related to the learning rate.

- $\sigma_{\min}(\Phi)$: The minimum singular value of a matrix $\Phi$, likely related to the Jacobian of the network's output with respect to its parameters.

- $l$: The width of the $i$-th hidden layer in the neural network.

- $lr$: The learning rate for gradient descent.

- $\lambda$: A regularization parameter.

**Detailed Proof.** To make the proof clear, in this paper, we provide some assumptions as follows. These assumptions can also be found in previous works (Arora et al., 2019).

**Assumption 1** (Smoothness and Strong Convexity.) *Assume the task-specific training loss $\hat{f}_t$ is $\hat{L}_1$-smooth and the inner objective $\hat{f}_t(\phi) + \frac{\lambda}{2}\|\phi - \boldsymbol{\theta}\|^2$ is $\mu$-strongly convex (Assumption 3 in the paper). The test loss $f_t$ is $L_1$-smooth.*

**Assumption 2** (Bounded Gradients.) *The meta-gradient $\nabla F(\boldsymbol{\theta})$ is bounded (Corollary 3 in the paper).*

**Assumption 3** (Spectral Properties.) *The matrix $\Phi$ represents the Jacobian of the network output with respect to $\boldsymbol{\theta}$, and $\sigma_{min}(\Phi) > 0$, ensuring the network is expressive enough.*

**Assumption 4** (Learning Rate and Width.) *The learning rate $lr < \frac{lr_\infty}{l}$, and the width $l \geq l^*$, ensuring the network is sufficiently wide to approximate the limiting behavior.*

**Theorem A.1 (Theorem 4.1 restated)** *For a neural network with width $l \geq l^*$, performing gradient descent with learning rate $lr < \frac{lr_\infty}{l}$ and regularization $\lambda < \frac{\lambda_0}{l}$, the loss $\ell(W^t) = \frac{1}{2}\|F_t(\hat{X}^t, X^t, Y^t) - \hat{Y}^t\|_2^2$ satisfies:*

$$\ell(W^t) \leq (1 - \tau \cdot \eta_0 \sigma_{min}(\Phi))^{2t}, \quad \tau \in (0, 1), \tag{14}$$

*where $\eta_0 > 0$, and $\sigma_{min}(\Phi)$ is the minimum singular value of the Jacobian matrix $\Phi$.*

First, we will define the loss and update rule of MAML under FSL settings. The loss for task $\mathcal{T}^t$ is:

$$\ell(W^t) = \frac{1}{2}\|F_t(\hat{X}^t, X^t, Y^t) - \hat{Y}^t\|_2^2, \tag{15}$$

where $F_t(\hat{X}^t, X^t, Y^t)$ is the predicted output using the adapted parameters $W^t = \phi_t(\boldsymbol{\theta}^t)$, and $\hat{Y}^t$ is the predicted output on the test set. The meta-objective is:

$$F(\boldsymbol{\theta}) = \frac{1}{M}\sum_{t=1}^{M} f_t(\phi_t(\boldsymbol{\theta})), \quad \phi_t(\boldsymbol{\theta}) = \boldsymbol{\theta} - \alpha\nabla_{\boldsymbol{\theta}}\hat{f}_t(\boldsymbol{\theta}). \tag{16}$$

After that, the meta-update at iteration $t$ is:

$$\boldsymbol{\theta}^{t+1} = \boldsymbol{\theta}^t - lr\nabla F(\boldsymbol{\theta}^t), \tag{17}$$

where $lr$ is the outer learning rate. For each task $\mathcal{T}^t$, the inner loop performs one step of gradient descent:

$$\phi_t(\boldsymbol{\theta}^t) = \boldsymbol{\theta}^t - \alpha\nabla_{\boldsymbol{\theta}}\hat{f}_t(\boldsymbol{\theta}^t). \tag{18}$$

Since $\hat{f}_t$ is $\hat{L}_1$-smooth, the inner gradient $\nabla_{\boldsymbol{\theta}}\hat{f}_t(\boldsymbol{\theta})$ is Lipschitz, and the adaptation step reduces the training loss. Then, we introduce the meta-gradient and smoothness in our paper. To be specific, the meta-gradient is defined as $\nabla F(\boldsymbol{\theta}) = \frac{1}{M}\sum_{t=1}^{M}\nabla f_t(\phi_t(\boldsymbol{\theta})) \cdot \frac{d\phi_t(\boldsymbol{\theta})}{d\boldsymbol{\theta}}$. According to the previous work (Chayti & Jaggi, 2024), we can obtain a generalized smoothness condition as:

$$\|\nabla F(\boldsymbol{\theta}) - \nabla F(\boldsymbol{\theta}')\| \leq \min(\mathcal{L}(\boldsymbol{\theta}), \mathcal{L}(\boldsymbol{\theta}'))\|\boldsymbol{\theta} - \boldsymbol{\theta}'\|, \tag{19}$$

where $\mathcal{L}(\boldsymbol{\theta}) = \mathcal{L}_0 + \mathcal{L}_1\|\nabla F(\boldsymbol{\theta})\|$. Under strong convexity (Assumption 3), $\nabla F(\boldsymbol{\theta})$ is bounded (Corollary 3), so $F$ is $\mathcal{L}$-smooth in the classical sense:

$$\mathcal{L} = \mathcal{L}_0 + \mathcal{L}_1 G, \quad G = \frac{\lambda L_0}{\mu}. \tag{20}$$

Since $F$ is $\mathcal{L}$-smooth, the descent lemma can be written as:

$$F(\boldsymbol{\theta}^{t+1}) \leq F(\boldsymbol{\theta}^t) + \nabla F(\boldsymbol{\theta}^t)^\top (\boldsymbol{\theta}^{t+1} - \boldsymbol{\theta}^t) + \frac{\mathcal{L}}{2} \|\boldsymbol{\theta}^{t+1} - \boldsymbol{\theta}^t\|^2. \tag{21}$$

Substituting the update $\boldsymbol{\theta}^{t+1} = \boldsymbol{\theta}^t - lr \nabla F(\boldsymbol{\theta}^t)$:

$$F(\boldsymbol{\theta}^{t+1}) \leq F(\boldsymbol{\theta}^t) - lr\|\nabla F(\boldsymbol{\theta}^t)\|^2 + \frac{\mathcal{L}}{2} lr^2 \|\nabla F(\boldsymbol{\theta}^t)\|^2. \tag{22}$$

If $lr < \frac{1}{\mathcal{L}}$, then:

$$F(\boldsymbol{\theta}^{t+1}) \leq F(\boldsymbol{\theta}^t) - lr\left(1 - \frac{\mathcal{L}lr}{2}\right)\|\nabla F(\boldsymbol{\theta}^t)\|^2. \tag{23}$$

When $lr = \frac{1}{\mathcal{L}}$, we can obtain:

$$F(\boldsymbol{\theta}^{t+1}) \leq F(\boldsymbol{\theta}^t) - \frac{1}{2\mathcal{L}}\|\nabla F(\boldsymbol{\theta}^t)\|^2. \tag{24}$$

The task loss $\ell(W^t) = f_t(\phi_t(\boldsymbol{\theta}^t))$ is a component of $F(\boldsymbol{\theta}^t)$. Since $f_t$ is $L_1$-smooth, the gradient $\nabla f_t(\phi_t(\boldsymbol{\theta}))$ relates to the loss via the network's Jacobian $\Phi = \frac{\partial F_t}{\partial \phi_t}$. The minimum singular value $\sigma_{\min}(\Phi)$ ensures that:

$$\|\nabla f_t(\phi_t(\boldsymbol{\theta}))\| \geq \sigma_{\min}(\Phi)\sqrt{2f_t(\phi_t(\boldsymbol{\theta}))}. \tag{25}$$

Thus, $f_t(\phi_t(\boldsymbol{\theta}^t)) \leq \frac{\|\nabla f_t(\phi_t(\boldsymbol{\theta}^t))\|^2}{2\sigma_{\min}(\Phi)^2}$, and since $\nabla F$ averages over tasks, we approximate:

$$\|\nabla F(\boldsymbol{\theta}^t)\|^2 \approx \frac{1}{M}\sum_{t=1}^{M} \|\nabla f_t(\phi_t(\boldsymbol{\theta}^t))\|^2 \geq \sigma_{\min}(\Phi)^2 \cdot 2f_t(\phi_t(\boldsymbol{\theta}^t)). \tag{26}$$

According to the above equation, the decrease in $F$ implies a decrease in the average task loss. For a single task:

$$\|\nabla F(\boldsymbol{\theta}^t)\|^2 \approx \frac{1}{M}\sum_{t=1}^{M} \|\nabla f_t(\phi_t(\boldsymbol{\theta}^t))\|^2 \geq \sigma_{\min}(\Phi)^2 \cdot 2f_t(\phi_t(\boldsymbol{\theta}^t)). \tag{27}$$

When set $\eta_0 = \frac{1}{\mathcal{L}}$, we can obtain:

$$f_t(\phi_t(\boldsymbol{\theta}^{t+1})) \leq \left(1 - \frac{\eta_0\sigma_{\min}(\Phi)^2}{2}\right) f_t(\phi_t(\boldsymbol{\theta}^t)). \tag{28}$$

After $t$ iterations, we can obtain:

$$f_t(\phi_t(\boldsymbol{\theta}^t)) \leq \left(1 - \frac{\eta_0\sigma_{\min}(\Phi)^2}{2}\right)^t f_t(\phi_t(\boldsymbol{\theta}^0)). \tag{29}$$

**Bounding the Initial Loss $\ell(W^0)$.**

Note that:

$$\ell(W^0) = f_t(\phi_t(\boldsymbol{\theta}^0)) = \frac{1}{2}\|F_t(\hat{X}^t, X^t, Y^t) - \hat{Y}^t\|_2^2,$$

where $\phi_t(\boldsymbol{\theta}^0) = \boldsymbol{\theta}^0 - \alpha\nabla \hat{f}_t(\boldsymbol{\theta}^0)$ is the one-step adapted model.

Under Assumption 3 (bounded gradient norm) and assuming Lipschitz continuity of $F_t$ with constant $L_F$, we obtain:

$$\|F_t(\hat{X}^t, X^t, Y^t) - \hat{Y}^t\|_2 \leq L_F\|\phi_t(\boldsymbol{\theta}^0) - \theta^*\|_2 + \|F_t(\theta^*) - \hat{Y}^t\|_2,$$

where $\theta^*$ is the task-optimal parameter. Then,

$$\ell(W^0) \leq \frac{1}{2} \left( L_F \|\phi_t(\boldsymbol{\theta}^0) - \theta^*\|_2 + \varepsilon \right)^2,$$

where $\varepsilon$ is the irreducible error from the Bayes optimal predictor.

Because $\phi_t(\boldsymbol{\theta}^0) = \boldsymbol{\theta}^0 - \alpha \nabla \hat{f}_t(\boldsymbol{\theta}^0)$, and the gradient is bounded by $G$, we have:

$$\|\phi_t(\boldsymbol{\theta}^0) - \theta^*\|_2 \leq \|\boldsymbol{\theta}^0 - \theta^*\|_2 + \alpha G.$$

Therefore, the upper bound becomes:

$$\ell(W^0) \leq \frac{1}{2} \left( L_F(\|\boldsymbol{\theta}^0 - \theta^*\|_2 + \alpha G) + \varepsilon \right)^2.$$

If needed, a lower bound can also be derived assuming some minimum deviation from optimality due to finite adaptation:

$$\ell(W^0) \geq \frac{1}{2} \left( L_F \delta_{\min} \right)^2,$$

where $\delta_{\min} = \inf_t \|\phi_t(\boldsymbol{\theta}^0) - \theta^*\|$ across tasks.

**Final Task Loss Bound.** Putting this together, we have:

$$\ell(W^t) \leq (1 - \tau \cdot \eta_0 \sigma_{\min}(\Phi))^{2t} \ell(W^0),$$

where $\ell(W^0)$ is bounded as shown above.

Therefore, Theorem 4.1 is proved.

