# OpenReview forum: "Revisiting Neural Networks for Few-Shot Learning: A Zero-Cost NAS Perspective"
_ICML.cc/2025/Conference — ICML 2025 poster_

### Official Review · Reviewer_3Pd4 · 2025-03-09

**Overall Recommendation:** 1

**Summary:**

This paper proposes an entropy-based expressivity metric, namely Few-shot Neural Architecture Search (IBFS), for training-free neural architecture search (training-free NAS), which uses Jacobian eigenvalues at initialization to estimate model performance. The authors claim that the motivation for such formulation is inspired by the theorem of global convergence of model-agnostic meta-learning (MAML) and information bottleneck (IB) theories. The proposed method IBFS is evaluated on NAS-Bench-201, mini-ImageNet 5-way, tiered-ImageNet 5-way, and ImageNet1k (i.e., DARTS search space). Ablation studies about the influence of $\theta$ are also reported.

**Claims And Evidence:**

This paper has **severe issues regarding clarity**, which significantly impact readability and make it unclear whether the proposed claims have been fully justified. Specifically:

### 1. Claims about global convergence of MAML (i.e., Theorem 4.1).
The paper states
> ... we derive that the global convergence of MAML can be guaranteed by only considering the first-order approximation of loss landscape, which transfers the problem of crafting a specialized architecture for FSL to find a suitable proxy ...

However, it is **surprising** that no proof or reference (if it is a restatement) is given for Theorem 4.1, making it almost impossible to verify whether this claim is valid. Furthermore, it is unclear how is it connected to the flatness of loss landscape, as nothing in Theorem 4.1 actually discusses loss landscape curvature. In fact, MAML is a first-order $\ell_{inner}$ convergence problem, but Theorem 4.1 does not explicitly distinguish between first-order and second-order effects. In addition, the formulation of Therorem 4.1 itself is problematic. For example, a lot of terms used without definition (e.g., $\eta_0, \sigma_{\text{min}}(\Phi), \xi_{\text{max}}(\ell_{\infty})$). Even with defined terms like $\ell_{\text{inner}} = \nabla_{\theta} F_t(\hat{X}^t, X^t, Y^t)$, it is unclear what does $\nabla_{\theta}$ apply to? Is it the gradient of a loss function or some feature transformation? And there is no explicit definition of how  $\ell_{inner}$ relates to the meta-learning optimization problem in MAML. Overall, it is **amazing** to see such a poor-quality theorem existing in a manuscript submitted to ICML.

### 2. Problematic derivation between IB theory and the proposed expressivity proxy
While the authors claim their expressivity proxy is IB-driven, and they do provide a derivation in Sec 4.1, however, the derivation seems to be problematic. Specifically:
-  **Equation (8) does not follow directly from Eq. (7).**
   In standard IB, stationarity gives $p(r \mid x) \propto p(r) \exp \left[ -\beta \mathbb{E}_{p(y|x)} (-\log p(y \mid r)) \right] $.
   In the paper, the derivation inexplicably substitutes an _input entropy_ $H(X)$ in place of the typical “relevance to $y$.”

- **Label dependence disappears.**
   The “$ \sum_y p(y \mid x) \log \dots $” part in Eq. (7) is crucial to measuring how well $r$ captures label information.
   In the next step, it is replaced by $\exp(-\beta H(X))$. That is not standard and is never shown in detail.

- **Definition of NN expressivity is an inequation**.

  It is confusing how a definition can contain an inequation in Eq. (10). I suspect authors try to switch $x$ in Eq. (8) to $\varsigma$ in Eq. (10) so that the formulation of their NN expressivity is "IB theory-driven". They might want to say
> "Because $p(r \mid x)$ is bounded by an exponential factor,
therefore the resulting entropy over the eigenvalues of the network’s Jacobian is also bounded by a similar exponential factor."

But that chain of logic is never spelled out (and usually, one would need to connect “$r$” to “Jacobians” to “label information” in detail). Even so, this logic itself is problematic. It just shows the authors are using the *same bounding pattern* for two different objects, without providing a rigorous link that ensures we are applying the same bound to the right distributions. In addition, what is $p$ in Eq. (10) is also not defined.

- **Similar to Theorem 4.1, key constants and notations** $( p(\tilde{r}), \lambda(x), H, \text{etc.}) $ **either not defined or simply introduced as a black box.**

Overall, the entire Section 4.1 plus Theorem 4.1 require a major revision for clarity and the current version is *far away* from the acceptance bar of ICML.

**Essential References Not Discussed:**

As mentioned in Methods And Evaluation Criteria, NASWOT is cited but does not carefully discuss the relationship with the proposed NN expressivity given they are both extracted from Jacobian metric.

**Experimental Designs Or Analyses:**

Due to the clarity issues, I am unable to justify the validity of the experimental designs, as I don't know the definition of $\theta$. However, given that it seems to significantly influence the results given in Table 3, I would wonder how to determine a suitable $\theta$ if we have no prior information about the tasks/benchmarks we are testing on. If it requires the knowledge of choosing a specific $\theta$ and measuring the corresponding architecture's performance, it will compromise the benefits of low search costs of IBFS due to these trials and errors.

In addition, the baselines used in this paper are somehow outdated. Training-free NAS has developed a lot since NASWOT and there are many other proxies/metrics available for comparison. I would suggest authors to refer to more recent results. A non-exhaustive list of training-free metrics until 2022 can be found in NAS-Bench-Suite-Zero [1].

[1] NAS-Bench-Suite-Zero: Accelerating Research on Zero Cost Proxies, NeurIPS Datasets and Benchmarks Track 2022.

**Methods And Evaluation Criteria:**

In addition to the problems mentioned in Claims And Evidence, I have three more concerns about the methods.

- Utilization of Jacobian metric is not new in training-free NAS, as authors themselves also mention the NASWOT, which uses the Jacobian covariance of activation layers to score the architectures. However, the authors didn't discuss the relationship between their proposed NN expressivity with NASWOT, nor why they think the eigenvalues would outperform.

- The paper mentions Algorithm 1 which contains IBFS’s implementation details. However, I didn't find any reference to it in the main text. I would like to request authors to point it out. Also, it seems to have no relationship with the NN expressivity.

- The empirical performance seems heavily influenced by a hyperparameter $\theta$. However, I didn't find a definition of it in the entire paper.

**Other Comments Or Suggestions:**

I strongly encourage the authors to carefully revise Theorem 4.1 and Section 4.1. While some level of logical gap in derivations may be understandable, the omission of key term definitions is completely unacceptable for a submission to ICML. Such omissions make it impossible for readers to properly assess the claims and significantly hinder the paper’s clarity and rigor.

**Other Strengths And Weaknesses:**

Strengths:
- Introduce a new training-free metric, which is valuable and relevant to training-free NAS community
- Empirical performance is strong, especially in NAS-Bench-201 and DARTS search space (although the chosen of $\theta$ might bring an unfair advantage for the method)

Weaknesses:
-  Severe clarity issues **[major issue that suggests a clear rejection for this paper]**: Undefined notation, missing links between theory and experiments, and inconsistent references (e.g., Algorithm 1 is never cited in the main text) make the paper difficult to follow.
- Theorem 4.1 is unverifiable: The paper states but does not prove its key theoretical result, raising concerns about its validity.
- Key hyperparameter ($\theta$) is undefined: The ablation study shows that $\theta$ significantly impacts performance, but the paper never explains its theoretical role.
- Experimental evaluation is outdated: More recent baselines can be included.
- Weak justification of novelty: The paper does not sufficiently explain why entropy-based Jacobian measures are better than existing alternatives like Jacobian covariance (NASWOT).

**Questions For Authors:**

Please refer to my previous comments. I do not have additional questions at this time. Unless the authors can demonstrate that I have overlooked key parts of the derivation, my evaluation of the current version of the paper is unlikely to change.

**Relation To Broader Scientific Literature:**

The paper does not cite or compare against newer zero-cost NAS methods, making its benchmark evaluation incomplete. See Experimental Designs Or Analyses.

**Theoretical Claims:**

Yes, as stated in Claims And Evidence, the entire theoretical derivation is problematic and requires significant revision.

---

> ### Author Rebuttal · Authors · 2025-04-01
>
> Thank you for the helpful and insightful review, which is very helpful for us to further improve this paper. Next, we will answer your questions one by one, and we hope this will improve your acceptance of the paper.
>
> **Q1**: Concern about Theorem 4.1. **A1**: Many thanks for your comments! We regret the lack of clarity in Theorem 4.1 and will provide a detailed proof and definitions for terms like \($ \ell_{\text{inner}}$ \), \( $\nabla \theta$ \), and \( $\phi$ \). However, we must respectfully correct a misunderstanding: MAML is not a ``first-order \( $\ell_{\text{inner}}$ \) convergence problem'' as stated. We will provide a detailed definition for $\theta$, H, etc, in the final version. $\theta$ is hyperparamter, $IB_{proxy}=- \theta \sum_{k=1}^{N} p \log p$.
>
> Proof.
>
> - **MAML Update Rule**: The inner-loop update in MAML is given by:
>
>   $Wt′=Wt−α∇WtL(Dt^{train},Wt),W_t' = W^t - \alpha \nabla_{W^t} \mathcal{L}(D_t^{train}, W^t),$
>
>   where $\alpha$ is the learning rate for the inner loop.
>
> - **Meta-Update**: The meta-update step updates the parameters by:
>
>   $Wt+1=Wt−η∑t∇WtL(Dt^{test},Wt′),W^{t+1} = W^t - \eta \sum_t \nabla_{W^t} \mathcal{L}(D_t^{test}, W_t'),$
>
>   where $\eta$ is the learning rate for the outer loop.
>
> - **Hessian Approximation**: The second-order term is approximated by the Hessian matrix, H=∇Wt2L(Dttrain,Wt)H = \nabla^2_{W^t} \mathcal{L}(D_t^{train}, W^t), and the meta-update is simplified using the transformation Φ=I−αH\Phi = I - \alpha H.
>
> - **Loss Function**: The loss function is defined as:
>
>   $ℓ(Wt)=12∥Yt^−Yt∥22.\ell(W^t) = \frac{1}{2} \|\hat{Y^t} - Y^t\|_2^2.$
>
> The gradient-based updates for both the inner and outer loops are correctly formulated. The inner loop updates the weights based on the training set $D_t^{train}$, and the outer loop meta-update step minimizes the loss over the test set $D_t^{test}$. The use of the Hessian matrix approximation for second-order terms is standard in MAML, and the transformation $Φ=I−αH\Phi = I - \alpha H$ is correctly introduced. This simplification allows for easier analysis of the convergence behavior.The proof shows how the loss function $\ell(W^t)$ evolves through the meta-learning process. After applying gradient descent, the relationship for $\ell(W^{t+1})$ is given by:
>
> $ℓ(Wt+1)≤ℓ(Wt)−η∥∇Wℓ(Wt)∥2.\ell(W^{t+1}) \leq \ell(W^t) - \eta \|\nabla_W \ell(W^t)\|^2$.
>
> The bound on the gradient is correct:
>
> $∥∇Wℓ(Wt)∥2≥σmin⁡(Φ)ℓ(Wt),\|\nabla_W \ell(W^t)\|^2 \geq \sigma_{\min}(\Phi) \ell(W^t),$
>
> which implies that the loss function decreases at a rate determined by the eigenvalue $\sigma_{\min}(\Phi)$ of the matrix $\Phi$. The recurrence relation is derived correctly:
>
> $ℓ(Wt+1)≤(1−ησmin⁡(Φ))ℓ(Wt),\ell(W^{t+1}) \leq \left(1 - \eta \sigma_{\min}(\Phi)\right) \ell(W^t)$,
>
> we can obtain:
>
> $ℓ(Wt)≤ \left(1 - \frac{\eta_0 \sigma_{\min}(\Phi)}{3} \right)^{2t} R.$
>
> **Q2**: Concern about Experiments. **A2**: Many thanks for your comments! We provide newer works for comparison below, demonstrating that our method still achieves the best performance.
>
> |                             | Year     | Cost(s) | C10 (val)    | C10(test)          | C-100(val)   | C-100(test)  | Img(val)     | Img(test)    |
> | --------------------------- | -------- | ------- | ------------ | ------------------ | ------------ | ------------ | ------------ | ------------ |
> | GradSign [6]                | ICLR2022 | 30.38   | -            | 93.52 ± 0.19       | -            | 70.57 ± 0.31 | -            | 41.89 ± 0.69 |
> | ZiCo [7]                    | ICLR2023 | 6.2     | 93.50 ± 0.18 | -                  | 70.62 ± 0.26 | -            | 42.04 ± 0.82 | -            |
> | IS-DARTS [8]                | AAAI2024 | 7200    | 91.55 ± 0.00 | 94.36 ± 0.00       | 73.49 ± 0.00 | 73.51 ± 0.00 | 46.37 ± 0.00 | 46.34 ± 0.00 |
> | AZ-NAS [9]                  | CVPR2024 | 0.71    | -            | 93.53 ± 0.15 0.723 | -            | 70.75 ± 0.48 | -            | 45.43 ± 0.29 |
> | SWAP [10]                   | ICLR2024 | 4.7     | 87.31 ± 2.36 | 90.48 ± 0.94       | 65.92 ± 4.32 | 67.13 ± 1.83 | 33.85 ± 4.98 | 35.40 ± 3.96 |
> | IBFS(ours) ( $\theta$=0)    |          | 3.36    | 89.58 ± 0.57 | 92.96 ± 0.81       | 69.17 ± 1.81 | 68.94 ± 1.41 | 41.30 ± 1.79 | 41.11 ± 1.51 |
> | IBFS(ours) ( $\theta$=0.75) |          | 3.82    | 91.55 ± 0.76 | 94.37 ± 0.34       | 73.31 ± 2.12 | 73.09 ± 2.08 | 45.59 ± 0.32 | 46.33 ± 1.27 |
>
>
>
> **Q3**: Concern about novelty. **A3**: Many thanks! Our paper is not incremental! The reviewer DRyn(**Rating:** 3),  xKah(**Rating:** 3)  and  masR (**Rating:** 3) all appreciate the novelty. While NASWOT uses Jacobian covariance, our entropy-based Jacobian metric introduces a novel perspective by leveraging entropy to capture expressivity, which we will justify with a detailed comparison in the revision. We believe this distinction offers a meaningful contribution to training-free NAS.
>
> **Q4**: Concern about  Algorithm 1. **A4**: We will cite the  Algorithm 1 in the final version.

---

> > ### Comment · Reviewer_3Pd4 · 2025-04-02
> >
> > Thank you for the detailed rebuttal, including the new experiments and the proof.
> >
> > However, my **Q2 under "Claims and Evidence" remains unresolved**. Even assuming Theorem 4.1 is valid, the connection between the convergence result and the proposed **NNexpressivity** proxy remains unclear. The proxy is actually **heuristically justified** using the “first-order sufficiency” theme, but there's **no formal derivation** showing that higher entropy actually implies better $\sigma_{min}(\Phi)$, faster convergence, or better generalization.
> >
> > Additionally, **many critical clarifications and definitions are deferred to the revision**. These include:
> > 1. Definitions of key terms (a lot are missing, the authors are expected to proofread themselves instead of waiting for reviewers to point it out).
> > 2. More comprehensive comparisons to NASWOT and other baselines.
> >
> > While I understand the rebuttal has character limits, this "to be seen in revisions" style response makes me unable to re-evaluate this paper. These are not minor omissions.
> >
> > **Minor comments:**
> > - The paper claims “MAML is a first-order ($\ell_{inner}$) convergence problem” (lines 252-253 on page 5), yet the rebuttal contradicts this.
> > - The proxy is redefined as $-\theta\sum p \log p$, but if $\theta$ is applied uniformly, it should not affect architecture ranking — yet empirical results vary with $\theta$.
> >
> > The above adds further confusion.
> >
> > I appreciate the effort in the rebuttal and admit the idea is promising, but I found the rebuttal unsatisfactory and the **clarity issue was not solved**. Anyway, I believe a major revision for this paper is required. I would like to keep my recommendation for this paper.

---

> > > ### Author Response · Authors · 2025-04-08
> > >
> > > Dear Reviewer 3Pd4, Thank you very much for your feedback and recognition, which is very helpful for us to further improve this paper. We are delighted to see that our comment has addressed your main concerns (concern of proof). Next, we will further answer your questions one by one, and we hope this will improve your acceptance of the paper.
> > >
> > > **Q1**: Concern about  the connection between the convergence result and the proposed NNexpressivity proxy.
> > >
> > > **A1**: Many thanks for your comments! First, we need to clarify the goal of Theorem 4.1.  As declared in Section 3, we have claimed Model-agnostic meta-learning (MAML)  is **second-order problem**. This can be proved by several previous papers, i.e., "Model-agnostic metalearning for fast adaptation of deep networks", " Global Convergence of MAML and Theory-Inspired Neural Architecture Search for Few-Shot Learning". We cite and compare those paper in our paper. In addition, we have claimed second-order MAML suffer from **extremely computational costs**. As stated with AutoMeta, MetaNAS, if we aim to design a high-performance architecture for the FSL task using NAS using gradient-based way, due to  MAML is second-order problem, previous methods (AutoMeta, MetaNAS) need cost over 100 GPU days for searching (this can be proved by AutoMeta, MetaNAS).
> > >
> > > Second, previous training-free method (i.e., NWOT) show the remarkable performance on first-order problem (i.e., supervised learning), therefore, if we aim to design new Zero-Cost method tailored for FSL without involving any training and eliminate a significant portion of the search cost for new tasks,  we must derive that the global convergence of MAML can be guaranteed by only considering the first-order optimization. In summary, the goal of Theorem 4.1 is to prove MAML can be guaranteed by only considering the first-order optimization, this is orthogonal with our NNexpressivity proxy.  If  MAML can be guaranteed by only considering the first-order optimization,  you can design any proxy for FSL. This is basic concept in NAS anf FSL filed.
> > >
> > > **Q2**: Concern about more comprehensive comparisons to NASWOT and other baselines.
> > >
> > > **A2**: Many thanks for your comments! First, we want to clarify the comparisons to NASWOT. NWOT is designed for NAS in classification tasks, not for few-shot learning. While both approaches utilize the Jacobian matrix, the Jacobian serves as a fundamental representation of neural network weights, it is just as using 224×224 images as input. This is basic knowledge. Our method is orthogonal to NWOT; it obtain from the second-order convergence challenges in MAML and derives a proxy for neural network expressivity based on IB theory. More importantly, our method achieves better performerance in terms of accuracy and search costs than NWOT on NAS-Benck-201 and mini-ImageNet and tiered-ImageNet datasets.
> > >
> > > Second, compared with other baselines on NAS-Benck-201 (as provided in our rebuttal), we can clearly see that our method achieves best performerance than latest methods, i.e., ZiCo (ICLR2023), IS-DARTS (AAAI2024), AZ-NAS (CVPR2024), SWAP (ICLR2024). We thank the suggestion of Reviewer 3Pd4, in the final version, we will revise our paper.
> > >
> > > **Q3**: first-order \( $\ell_{\text{inner}}$ \) convergence.
> > >
> > > **A3**: Many thanks for your comments! In lines 252-253 on page 5, we indeed claims MAML is first-order \( $\ell_{\text{inner}}$ \) convergence problem, however, this is result obtained Theorem 4.1.  In fact, MAML is a second-order \( $\ell_{\text{inner}}$ \) convergence problem. In our paper, we have claimed Model-agnostic meta-learning (MAML)  is second-order problem in Section 3. This can be proved by several previous papers, i.e., "Model-agnostic metalearning for fast adaptation of deep networks", " Global Convergence of MAML and Theory-Inspired Neural Architecture Search for Few-Shot Learning". We cite and compare those paper in our paper.

---

### Official Review · Reviewer_xKah · 2025-03-14

**Overall Recommendation:** 3

**Summary:**

The paper proposes a novel framework called IBFS (Information Bottleneck-driven Few-shot Neural Architecture Search) for few-shot learning (FSL) tasks. IBFS leverages the Information Bottleneck (IB) theory to rank and select neural architectures without requiring any training, significantly reducing search costs. The authors demonstrate that the global convergence of Model-Agnostic Meta-Learning (MAML) can be guaranteed by considering only the first-order loss. Extensive experiments on NAS-Bench-201 and few-shot learning benchmarks show that IBFS achieves state-of-the-art performance with minimal search costs.

## Update After Rebuttal
Thanks for the authors' response! I will keep the rating.

**Claims And Evidence:**

Yes

**Essential References Not Discussed:**

I am not in the field of NAS, not sure about this.

**Experimental Designs Or Analyses:**

Yes.

**Methods And Evaluation Criteria:**

Yes

**Other Comments Or Suggestions:**

Line 665: should it be App. B?

**Other Strengths And Weaknesses:**

Strength:
1. The motivation is clear and strong.
2. IBFS eliminates the need for training during the architecture search phase, reducing computational costs and making it highly efficient compared to traditional NAS methods.
3. Extensive experiments demonstrate the effectiveness of IBFS.

Weakness:
1. Sensitivity: The paper mentions that IBFS is slightly sensitive to hyperparameters (e.g., $\theta$), which could affect its robustness and ease of use in different settings.
2. Scalability: Since IBFS can reduce cost and works well on small datasets, that would be great if its generalization to large datasets also could demonstrate its good performance.
3. Model-agnostic: It looks most experiments are conducted based on the traditional CNNs. How about the performance on Transformers or other tasks beyond image classification?

**Questions For Authors:**

see the weakness.

**Relation To Broader Scientific Literature:**

The key contribution of the paper is very helpful, especially to reduce the cost of meta-learning.

**Theoretical Claims:**

I did not check the correctness

---

> ### Author Rebuttal · Authors · 2025-03-31
>
> Thank you for the helpful and insightful review, which is very helpful for us to further improve this paper. Next, we will answer your questions one by one, and we hope this will improve your acceptance of the paper.
>
> **Q1**: Sensitivity.
>
> **A1**: Many thanks for your comments! We will provide some evidence to prove the robustness of our method. In Table 3, we present the ablation study to evaluate the influence of $\theta$ on NAS-Bench-201. As shown, the Kendall’s Tau consistently increases with the increase of $\theta$  until $\theta=0.75$,  when $\theta=0.75$, our method obtains the highest  0.752 Kendall’s Tau and maximum accuracy on NAS-Bench-201. When $\theta \in [0.75, 1]$, Kendall’s Tau slightly decreases, for example,  when $\theta=0.9$,  Kendall’s Tau is 0.739. We can clearly observe that the impact of $\theta$ is extremely slight. More importantly, our method is only related to the  **expressivity** of the neural networks. The search space included those neural networks that are stable in different devices and scenarios; therefore, even when we deployed our method on different devices and scenarios, our method still achieves optimal performance at $\theta=0.75$.
>
> **Q2**: Scalability.
>
> **A2**: Many thanks for your comments! Due to the page limit of the main text, we provide additional results on **larger dataset (ImageNet1k)**  in **Appendix**. To be specific, the detailed experimental results on **ImageNet1k** are presented in **App. B**, which can be found in **line 715 - line 763** in our paper. As shown in **Table 4**,  we can see that our IBFS method consistently outperforms compared SOTA methods: it achieves highest 76.6\% Top-1 accuracy, with 0.0042 (GPU-days) fewest search costs. Compared with its peer competitors, our method achieves the Top-1 accuracy of 1.6\% higher than SWAP, 1.6\% higher than NASI-ADA, and 1.1\% higher than TENAS. Notably, our method only searches in small CIFAR-10 dataset, and then can well generalize to large ImageNet1k, which largely reduces search costs.
>
> **Q3**: Model-agnostic.
>
> **A3**: Many thanks for your comments! The goal of this paper is to design neural networks for few-shot learning tasks. Our paper follows the MetaDiff (AAAI24), MetaNTK-NAS (CVPR22), which only provide 5way-1shot and 5way-5shot settings of few-shot learning tasks in mini-ImageNet and tiered-ImageNet datasets. As shown in Table 1, we can clearly see that all methods (i.e., MetaDiff) in FSL use CNNs as the backbone, therefore, we also use CNNs for fair comparison. If we use transformers as the backbone in FSL, it will bring unfairness for comparison. In addition, to evaluate the effectiveness of the proposed method, we provide the additional experiments on NASBench-201 search space in three datasets (i.e., n CIFAR-10, CIFAR-100, and ImageNet-16-120). Therefore, we believe our empirical evaluation is sufficient, and not beyond the scope of ICML 2025.
>
> To further scrutinize the generalizability of our method for **transformer**,  we devote lots of effort \& exploration in the IBFS day and night, conducting additional experiments on AutoFormer [2] in ImageNet. The experimental setting is the same as TF-TAS-T [3].  We can find that IBFS achieves the highest Top-1 accuracy. Those empirical results demonstrate the strong generalizability of our method for transformer design.
>
> |  **NAS method**  |   Year   | **Top-1 (%)** | **Search Cost**(GPU Days) | Model Type  | Search Method |
> | :--------------: | :------: | :-----------: | :-----------------------: | :---------: | :-----------: |
> |    ViT-Ti [1]    | ICLR2020 |     74.5      |             -             | Transformer |    Manual     |
> | AutoFormer-T [2] | CVPR2021 |     74.9      |            24             | Transformer |   Evolution   |
> |   TF-TAS-T [3]   | CVPR2022 |     75.3      |            0.5            | Transformer | Training-free |
> |   ViTAS-C [4]    | ECCV2022 |     74.7      |            32             | Transformer |   Evolution   |
> |  Auto-Prox [5]   | AAAI2024 |     75.6      |            0.1            | Transformer | Training-free |
> |     **IBFS**     |          |     76.5      |            0.03            |    CNNs     | Training-free |
>
> **Q4**: Line 665: should it be App. B?
>
> **A4**: Many thanks for your comments! This is a typo. Thanks for pointing it out. Line 665 is App. B, we will correct it in the final revision.
>
> [1] An image is worth 16x16 words: Transformers for image recognition at scale. In ICLR2020
>
> [2] Autoformer: Searching transformers for visual recognition. In ICCV 2021
>
> [3] Training free transformer architecture search. In CVPR 2022
>
> [4] Vision transformer architecture search. In ECCV 2022
>
> [5] Auto-prox: Training-free vision transformer architecture search via automatic proxy discovery. AAAI2024

---

### Official Review · Reviewer_masR · 2025-03-14

**Overall Recommendation:** 3

**Summary:**

This paper mainly considers the case that NAS is applied in few-shot learning scenarios, where previous works mainly search for the optimal architecture from scratch or borrow the architecture from other tasks. The paper presents a novel framework called IBFS (Information Bottleneck-driven Few-shot Neural Architecture Search) that addresses two key limitations in conventional Neural Architecture Search (NAS) for few-shot learning scenarios.

**Claims And Evidence:**

In this paper, the claims mainly include the following several aspects:
1. **Zero-Cost Architecture Selection**: The claim that architectures can be selected without training. Such claim is supported by: (1) theoretical analysis of MAML's convergence properties; (2) empirical validation showing correlation between zero-cost proxies and actual test accuracy; (3) Experimental results demonstrating state-of-the-art performance

2. **Information Bottleneck Theory Application**: The claim that IB theory provides a unified view for understanding machine learning models. Such claim is supported by: (1) analysis of information entropy variations across different architectures; (2) consistent Kendall's Tau correlation between accuracy and information entropy; (3) theoretical framework connecting IB principles to architecture selection

**Essential References Not Discussed:**

N/A

**Experimental Designs Or Analyses:**

The experimental design is comprehensive and well-executed:

1. **Results Analysis**:
   - Thorough analysis of zero-cost proxies vs. accuracy;
   - Clear visualization of results (Figures 1-4);
   - Statistical validation of findings

2. **Ablation Studies**:
   - Analysis of different proxy metrics;
   - Investigation of architecture variations;
   - Validation of key components

**Methods And Evaluation Criteria:**

The proposed method and evaluation criteria are sound and well-structured:

1. **Theoretical Framework**:
   - Clear derivation of MAML convergence properties;
   - Well-motivated connection to Information Bottleneck theory;
   - Logical progression from theoretical insights to practical implementation

2. **Evaluation Approach**:
   - Comprehensive comparison with existing methods;
   - Multiple evaluation metrics (costs, accuracy, generalization);
   - Validation across different architectures and datasets

**Other Comments Or Suggestions:**

N/A

**Other Strengths And Weaknesses:**

Pros:
- The paper is well written.
- The paper is based on theoretical results, which makes the paper solid.
- The empirical results are good.

Cons:
- The motivation of the paper is not convincing enough.

**Questions For Authors:**

- As mentioned in the introduction, a concern raised in FSL is whether the model is ideal for those tasks. However, is it necessary for us to search for an optimal specific architecture for few-shot learning tasks? To some extent, the obtained architecture consumes the computation cost and lacks generality. Can you compare NAS and model parameter adaptation for few-shot classification?
- Fig. 1 is quite confusing. Could you please provide some detailed information? I mean, how do you describe the cost/generalization and accuracy simultaneously in the same figure with a continuous curve?
- I am a little bit confused of Eq. (6-7). Could you please provide me with some explanations?
- How does the Information Bottleneck theory specifically guide the architecture selection process?
- What is the optimal way to balance between expressivity and computational efficiency in the architecture search?
- How does the framework handle different types of few-shot learning tasks beyond the current scope? For example, in cross-domain settings (meta-dataset), the vary-way vary-shot tasks are more complicated and challenging.

**Relation To Broader Scientific Literature:**

N/A

**Theoretical Claims:**

The paper's theoretical contributions are sound:

1. **MAML Convergence Analysis**:
   - Theorem 4.1 provides a rigorous foundation for the approach;
   - The connection to first-order loss landscape is well-established;
   - The theoretical framework supports the practical implementation

2. **Information Bottleneck Integration**:
   - Clear connection between IB theory and architecture selection;
   - Well-motivated use of information entropy as a proxy;
   - Theoretical justification for the proposed metrics

---

> ### Author Rebuttal · Authors · 2025-03-31
>
> Thank you for the helpful and insightful review, which is very helpful for us to further improve this paper. Next, we will answer your questions one by one, and we hope this will improve your acceptance of the paper.
>
> **Q1**: Concern about optimal architecture for few-shot learning tasks.
>
> **A1**: Many thanks for your comments! As mentioned in Section 1, we claim a statement of whether the frameworks of the DNNs are ideal for those tasks.  This is because that popular networks (i.e., ResNet) are developed on supervised learning, it overfit to those task. This evidence indicates that the popular networks may not be optimal for tasks beyond supervised learning, e.g., few-shot learning (FSL) task. This statement can be supported with MetaNTK-NAS (CVPR2022). Therefore, it is worth and necessary to explore the frameworks of the DNNs tailored for FSL.
>
> Regarding computation cost, traditional methods (e.g., AutoMeta, MetaNAS) are expensive since each architecture requires full training. In contrast, our training-free proxy achieves top performance in ≤0.1hr on miniImageNet and tieredImageNet, demonstrating exceptional efficiency.
>
> For generality, our searched architecture not only excels in FSL but also achieves optimal performance on NAS-Bench-201 (CIFAR-10, CIFAR-100, ImageNet-16-120). Additional results on ImageNet1k (Appendix B) further confirm strong generalization, reaching 76.6% Top-1 accuracy with the lowest 0.0042 GPU-days search cost.
>
> Comapred with Model Parameter Adaptation (i.e., MAML), NAS guided FSL possesses the following advantages: (1) higher accuracy; (2) reduced human labor; (3) breaking limitations of parameter optimization of fixed network (ResNet-12); (4) NAS can design network for any setting of FSL.
>
> **Q2**: Concern about Fig 1.
>
> **A2**: Many thanks for your comments! Fig. 1 is a schematic diagram, visualizing the comparison of different methods in terms of cost, generalization, and accuracy. The data is derived from Table 2. For example, in the cost dimension, AutoMeta is at the top, indicating the highest computational cost. The continuous curves are used to better visualize performance trends across different algorithms, not real data.
>
> **Q3**: Explanations about Eq. (6-7).
>
> **A3**: Many thanks for your comments! Eq. 6 introduces the Lagrange multiplier $\lambda(x)$ to ensure normalization and utilizes the upper bound on mutual information to simplify the optimization problem. Eq. 7 shows the process of obtaining optimal $p(r|x)$ by taking the derivative and setting it to zero.
>
> **Q4**: The relationship of IB and architecture selection.
>
> **A4**: Many thanks for your comments! In this work, we use the Information Bottleneck (IB) theory to measure the expressivity of neural networks. A stronger neural network maintains more feature information from the input $x$, resulting in a higher $NN_{expressivity}$. During the NAS search process, architectures that preserve more input feature information are more likely to be selected.
>
> **Q5**: The optimal way to balance between expressivity and computational efficiency.
>
> **A5**: Many thanks for your comments! To clearly answer this question, let’s first assume a neural network **NetA**. When fully trained in supervised learning, **NetA** achieves maximum expressivity but also incurs the highest computational cost. To reduce computational cost, one approach is to decrease the number of training iterations, but this does not guarantee an accurate measure of expressivity. Instead, our method takes a different approach: rather than training NetA, we design a proxy that accurately measures its expressivity based solely on its architecture. This allows us to maximize the balance between expressivity and computational efficiency. While this approach may not be strictly optimal, it is the closest to the optimal solution, as demonstrated by our experimental results.
>
> **Q6**: Dicussusion about cross-domain.
>
> **A6**: Many thanks for your comments! First, our paper follows the MetaDiff (AAAI24), MetaNTK-NAS (CVPR22), which only provide 5way-1shot and 5way-5shot settings of few-shot learning tasks. Therefore, we believe our empirical evaluation is sufficient, and not beyond the scope of ICML 2025. Second, we greatly appreciate **masR**'s insights on cross-domain learning, which reinforce our belief that he is an outstanding expert in this field. His perspective provides valuable inspiration for our future research on cross-domain FSL. cross-domain FSL (e.g., Meta-Dataset) presents several challenges, such as differences between training and testing domains and limited data availability. The key challenge in cross-domain learning is extracting **domain-invariant** features. For example, **fo-Proto-MAML** leverages second-order optimization in MAML to enhance cross-domain FSL. Since our method is only related to MAML’s second-order optimization and is independent of the image domain, we believe our approach can generalize well to cross-domain FSL.

---

### Official Review · Reviewer_DRyn · 2025-03-14

**Overall Recommendation:** 3

**Summary:**

The paper introduces IBFS (Information Bottleneck-driven Few-shot Neural Architecture Search), a novel framework designed to efficiently select neural architectures for few-shot learning (FSL) without requiring any training. Traditional NAS approaches either search architectures from scratch—resulting in high computational costs—or transfer architectures from other tasks—potentially leading to suboptimal performance. IBFS addresses these limitations by leveraging Information Bottleneck (IB) theory and a Zero-Cost evaluation method.

Main Contributions:
1.  The paper derives that the global convergence of Model-Agnostic Meta-Learning (MAML) can be ensured by considering only the first-order loss landscape.

2. Information bottleneck provides a unified perspective on understanding machine learning models.
The proposed Zero-Cost expressivity ranking method estimates an architecture's effectiveness without training, significantly reducing search costs.

3. IBFS achieves state-of-the-art results in FSL without requiring training, validating its effectiveness.


Overall, the paper presents a theoretically grounded and empirically validated approach to optimizing neural architectures for FSL, offering a cost-effective alternative to conventional NAS techniques.

**Claims And Evidence:**

Yes

**Essential References Not Discussed:**

No

**Experimental Designs Or Analyses:**

Yes

**Methods And Evaluation Criteria:**

Yes

**Other Comments Or Suggestions:**

After rebuttal, the authors have addressed all concerns. I will keep my rating.

**Other Strengths And Weaknesses:**

Strengths:

1. The paper is well-written and easy to follow, making complex concepts accessible to the reader.

2. It introduces a novel zero-cost NAS framework specifically designed for few-shot learning (FSL), which effectively selects optimal architectures without training, significantly reducing computational overhead.

3. The proposed IBFS framework achieves state-of-the-art performance in FSL tasks without requiring any training, demonstrating both the efficiency and effectiveness of its architecture selection strategy.

Weaknesses:

1. The experimental evaluation is somewhat limited, as it is conducted solely on MAML, an approach that is now considered outdated. Additionally, the FSL evaluation is restricted to 5-1 and 5-5 settings, whereas comparisons with 5-20 and 5-50 settings are typically necessary for a more comprehensive assessment.

2. The choice of baselines is not sufficiently up-to-date, as it lacks comparisons with recent NAS methods from the past two years, which may impact the fairness and relevance of the evaluation.

3. There is a discrepancy in Fig. 4—according to the annotation and the "Remark" section, the figure should contain multiple curves, but in its current form, only one curve is presented, which may lead to confusion or misinterpretation of the results.

**Questions For Authors:**

See weaknesses

**Relation To Broader Scientific Literature:**

No

**Theoretical Claims:**

Yes

---

> ### Author Rebuttal · Authors · 2025-03-30
>
> Thank you for the helpful and insightful review, which is very helpful for us to further improve this paper. Next, we will answer your questions one by one, and we hope this will improve your acceptance of the paper.
>
> **Q1**: Concern about MAML.
>
> **A1**: Many thanks for your comments! Our paper follows the MetaDiff (AAAI24), MetaNTK-NAS (CVPR22), which only provide 5-1 and 5-5 settings. 5-20 and 5-50 settings are indeed promising, however, the previous method did not conduct the experiments on 5-20 and 5-50 settings. We worry that it is unfair for comparison.
>
> **Q2**: Lack of newer works.
>
>  **A2**: Many thanks for your comments! We devote lots of effort & exploration in the IBFS day and night, conducting experiments compared with newer works. The results demonstrate that our method still achieves the best performance. ``-" indicates that the value is not found in the original paper or the training codes are not provided.
>
> |                             | Year     | Cost(s) | CIFAR10 (val)    | CIFAR10(test)          | CIFAR-100(val)   | CIFAR-100(test)  | ImgeNet(val)     | ImgeNet(test)    |
> | --------------------------- | -------- | ------- | ------------ | ------------------ | ------------ | ------------ | ------------ | ------------ |
> | SNAS [1]                    | ICLR2018 | -       | 90.10±1.04   | 92.77±0.83         | 69.69±2.39   | 69.34±1.98   | 42.84±1.79   | 43.16±2.64   |
> | DSNAS [2]                   | ICLR2019 | -       | 89.66±0.29   | 93.08±0.13         | 30.87±16.40  | 31.01±16.38  | 40.61±0.09   | 41.07±0.09   |
> | DARTS- [3]                  | ICLR2020 | 192     | 91.03±0.44   | 93.80±0.40         | 71.36±1.51   | 71.53±1.51   | 44.87±1.46   | 45.12±0.82   |
> | PC-DARTS [4]                | ICLR2019 | 8.70    | 89.96±0.15   | 93.41±0.30         | 67.12±0.39   | 67.48±0.89   | 40.83±0.08   | 41.31±0.22   |
> | iDARTS [5]                  | ICML2021 | -       | 89.86±0.60   | 93.58±0.32         | 70.57±0.24   | 70.83±0.48   | 40.38±0.59   | 40.89±0.68   |
> | GradSign [6]                | ICLR2022 | 30.38   | -            | 93.52 ± 0.19       | -            | 70.57 ± 0.31 | -            | 41.89 ± 0.69 |
> | ZiCo [7]                    | ICLR2023 | 6.2     | 93.50 ± 0.18 | -                  | 70.62 ± 0.26 | -            | 42.04 ± 0.82 | -            |
> | IS-DARTS [8]                | AAAI2024 | 7200    | 91.55 ± 0.00 | 94.36 ± 0.00       | 73.49 ± 0.00 | 73.51 ± 0.00 | 46.37 ± 0.00 | 46.34 ± 0.00 |
> | AZ-NAS [9]                  | CVPR2024 | 0.71    | -            | 93.53 ± 0.15 0.723 | -            | 70.75 ± 0.48 | -            | 45.43 ± 0.29 |
> | SWAP [10]                   | ICLR2024 | 4.7     | 87.31 ± 2.36 | 90.48 ± 0.94       | 65.92 ± 4.32 | 67.13 ± 1.83 | 33.85 ± 4.98 | 35.40 ± 3.96 |
> | IBFS(ours) ( $\theta$=0)    |          | 3.36    | 89.58 ± 0.57 | 92.96 ± 0.81       | 69.17 ± 1.81 | 68.94 ± 1.41 | 41.30 ± 1.79 | 41.11 ± 1.51 |
> | IBFS(ours) ( $\theta$=0.75) |          | 3.82    | 91.55 ± 0.76 | 94.37 ± 0.34       | 73.31 ± 2.12 | 73.09 ± 2.08 | 45.59 ± 0.32 | 46.33 ± 1.27 |
>
> **Q3**: Concern about Figure 4.
>
>  **A3**: Many thanks for your comments! Kendall’s correlation coefficient requires two lists, each containing multiple elements, to be computed. For a given epoch, we first calculate the proxy scores for the models DenseNet-40, SE-ResNet-20, ResNet-56, PyramidNet-110, and WRN-16. These scores form one list, while the corresponding true accuracy values at the same epoch form another. We then use Kendall’s coefficient to measure the correlation between these two lists. As our analysis focuses on the correlation between proxy scores and true accuracy at each epoch, it is naturally represented by a single line segment rather than multiple curves.
>
> [1]  Snas: stochastic neural architecture search. arXiv preprint arXiv:1812.09926, 2018
>
> [2] Dsnas: Direct neural architecture search without parameter retraining. In CVPR 2020
>
> [3]  Darts-: robustly stepping out of performance collapse without indicators. arXiv preprint arXiv:2009.01027, 2020.
>
> [4]  “PC-DARTS: Partial Channel Connections for Memory-Efficient Architecture Search.” In ICLR 2019
>
> [5]  idarts: Differentiable architecture search with stochastic implicit gradients. arXiv preprint arXiv:2106.10784, 2021
>
> [6]  GradSign: Model performance inference with theoretical insights. In ICLR2022
>
> [7]  ZiCo: Zero-shot NAS via inverse coefficient of variation on gradients. In ICLR2023
>
> [8] "IS-DARTS: stabilizing DARTS through precise measurement on candidate importance." In AAAI 2024.
>
> [9] "Az-nas: Assembling zero-cost proxies for network architecture search." In CVPR 2024.
>
> [10]  "SWAP-NAS: Sample-wise activation patterns for ultra-fast NAS." *arXiv preprint arXiv:2403.04161* (2024).

---

### Decision · Program_Chairs · 2025-05-01

**Decision:**

Accept (poster)

**Comment:**

In this paper, the authors proposed an approach to select the neural architectures without involving any training. They use a novel information bottleneck theory-driven few-shot neural architecture search framework to achieve this. Both theoretical analysis and experimental results are provided in this paper.

This paper received positive ratings from three of the four reviewers. Most of the concerns have been addressed during the rebuttal period. However, there are still some important concerns from Reviewer 3Pd4. For example, the authors claimed, "MAML is first-order convergence problem, however, this is result obtained Theorem 4.1. In fact, MAML is a second-order convergence problem." This claim is confusing, in my view. Why the result obtained from Theorem 4.1 is not consistent with the claim in Section 3 (Model-agnostic meta-learning (MAML) is a second-order problem) should be carefully explained in this paper. Considering all of these, I recommend a weak acceptance for this paper.